

# Regional modelling of polycyclic aromatic hydrocarbons: WRF/Chem-PAH model development and East Asia case studies

Qing Mu[1,2], Gerhard Lammel[1,2], Christian N. Gencarelli[3], Ian M. Hedgecock[3], Ying Chen[1,4], Petra Přibylová[2], Monique Teich[4], Yuxuan Zhang[1], Guangjie Zheng[5], Dominik van Pinxteren[4], Qiang Zhang[6], Hartmut Herrmann[4], Manabu Shiraiwa[1,7], Peter Spichtinger[8], Hang Su[1,9], Ulrich Pöschl[1], Yafang Cheng[1,9]

[1]Multiphase Chemistry Department, Max Planck Institute for Chemistry, Mainz, Germany
[2]Research Centre for Toxic Compounds in the Environment, Masaryk University, Brno, Czech Republic
[3]CNR-Institute of Atmospheric Pollution Research, Division of Rende, Rende, Italy
[4]Leibniz Institute for Tropospheric Research, Leipzig, Germany
[5]Brookhaven National Laboratory, Brookhaven, USA
[6]Ministry of Education Key Laboratory for Earth System Modeling, Department of Earth System Science, Tsinghua University, Beijing, China
[7]Department of Chemistry, University of California, Irvine, USA
[8]Institute for Atmospheric Physics, Johannes Gutenberg University, Mainz, Germany
[9]Institute for Environmental and Climate Research, Jinan University, Guangzhou, China

*Correspondence to: Y. F. Cheng (yafang.cheng@mpic.de) or G. Lammel (g.lammel@mpic.de)*

**Abstract.** Polycyclic aromatic hydrocarbons (PAHs) are hazardous pollutants, with increasing emissions in pace with economic development in East Asia, but their distribution and fate in the atmosphere have not yet been well understood. We extended the regional atmospheric chemistry model WRF/Chem (Weather Research Forecast model with Chemistry module) to comprehensively study the atmospheric distribution and the fate of low concentrated, slowly degrading semivolatile compounds. The WRF/Chem-PAH model reflects the state-of-the-art understanding of current PAHs studies with several new or updated features. It was applied for PAHs covering a wide range of volatility and hydrophobicity i.e., phenanthrene, chrysene and benzo(a)pyrene, in East Asia. Temporally highly resolved PAH concentrations and particulate mass fractions were evaluated against observations. The WRF/Chem-PAH model is able to reasonably well simulate the concentration levels and particulate mass fractions of PAHs near the sources and at a remote outflow region of East Asia, in high spatial and temporal resolutions. Sensitivity study shows that the heterogeneous reaction with ozone and the homogeneous reaction with the nitrate radical significantly influence the fate and distributions of PAHs. The methods to implement new tracers and to correct the transport problems can be applied to other newly implemented tracers in WRF/Chem.

## 1 Introduction

Polycyclic aromatic hydrocarbons (PAHs), released into the atmosphere as by product of all kinds of combustion processes, are harmful for human health via inhalation as well as ingestion pathways (World Health Organization, 2003; Lv et al., 2015) and for ecosystems (Hylland, 2006). In the atmospheric environment PAHs are partly readily degradable, partly undergo long-range transport and reach remote areas (Keyte et al., 2013). PAHs have been



included in the United Nations Economic Commission for Europe Convention on Long-range Transboundary Air Pollution and Protocol on Persistent Organic Pollutants. Hazardous substances, mostly benzo(a)pyrene (BaP), are criteria pollutants in many countries, including the European Union, USA and Japan. The United States Environmental Protection Agency (USEPA) prioritized 16 PAHs in the 1970s, which have been mostly targeted in the environment since then, but this selection is questionable considering toxicity and occurrence of PAHs (Andersson and Achten, 2015).

As PAHs are mainly generated from incomplete combustion of carbonaceous bio- and fossil fuels, their emissions increase dramatically in Asia due to rapid economic development and energy consumption. According to Zhang and Tao (2009), the total emission of the sixteen PAH compounds listed in the USEPA priority control list was about 290 Gg yr$^{-1}$ in Asia in year 2004, accounting for more than half of total global emissions. High emissions of PAHs in Asia pose a hazard to the ecosystems and human health on an intercontinental or even global scale (Hung et al., 2005).

In an attempt to elucidate the spatiotemporal distributions of PAH ambient concentrations and processes governing their atmospheric fate, several numerical modelling studies have been published. Lagrangian frameworks have been used for Europe (van Jaarsveld et al., 1997; Halsall et al., 2001) and China (Lang et al., 2007; Lang et al., 2008). Other studies focused on the multicompartmental behaviour using box models (Yaffe et al., 2001; Prevedouros et al., 2004; Prevedouros et al., 2008). Eulerian chemical transport models have been developed and applied for regions, i.e. Europe (Aulinger et al., 2007; Matthias et al., 2009; Bieser et al., 2012; San Jose et al., 2013; Efstathiou et al., 2016), North America (Galarneau et al., 2014) and East Asia (Zhang et al., 2009; Zhang et al., 2011a; Zhang et al., 2011b; Inomata et al., 2012; Inomata et al., 2013), or on the global scale (Sehili and Lammel, 2007; Lammel et al., 2009; Friedman and Selin, 2012; Friedman et al., 2014a; Friedman et al., 2014b; Shen et al., 2014; Shrivastava et al., 2017). The aforementioned studies differ in many respects relating to the PAH species examined, the temporal variability of their emissions, the spatial and process resolutions of the models.

Since its initial release in 2002, Weather Research Forecast model with Chemistry module (WRF/Chem) has been widely applied and verified for regional air quality (Zhang et al., 2010; Zhang et al., 2013) and climate (Liao et al., 2014; Yahya et al., 2015) study with high temporal and spatial resolutions. Comparing to previous PAH modelling studies, this work is unique in four aspects: (1) to include all relevant state-of-the-art processes of PAH into WRF/Chem which are important for its cycling in the atmospheric environment over land (i.e., new heterogeneous degradation scheme, several oxidants in homogeneous degradation processes, and re-volatilisation from soil, among others), (2) to predict and validate against observed gas and particulate phase PAH concentrations separately, (3) to validate atmospheric concentrations and particulate mass fraction against diurnal variable PAH observations, and (4) to explore the significance of PAH heterogeneous reaction with ozone (O$_3$) and gas-phase reaction with nitrate radical (NO$_3$).



## 2 Model development

WRF/Chem-PAH is based on the open source community model WRF/Chem (version 3.6.1), which is a fully coupled, "online" regional model with integrated meteorological, gas-phase chemistry, and aerosol components (Grell et al., 2005). It is built on the Advanced Research WRF core, which handles the dynamics, physics, and

transport processes. Gas-phase chemistry and aerosol schemes are integrated over the same time step as transport processes, allowing for full coupling between the schemes. The short chemistry time step also made the model ideal for studying short-lived PAH species with high levels of spatial heterogeneity.

By considering the compatibility with the WRF/Chem Kinetic PreProcessor (KPP) (Sandu et al., 2003; Sandu and Sander, 2006) and similarities of PAHs chemistry to volatile organic aerosol formations, we have made the following

choices. The Regional Atmospheric Chemistry Mechanism (RACM) (Stockwell et al., 1997) is used for homogeneous gas-phase reactions. The aerosol module includes the Modal Aerosol Dynamics Model for Europe (MADE) (Ackermann et al., 1998) for the inorganic fraction, and the Secondary Organic Aerosol Model (SORGAM) (Schell et al., 2001) for the secondary organic aerosols. MADE/SORGAM in WRF/Chem uses the modal approach with three log-normally distributed modes (nuclei, accumulation and coarse mode). All pollutant species normally

simulated in the standard RACM/SORGAM mechanism are also simulated in WRF/Chem-PAH. In order to include PAHs (and organics in general) in air-soil gas exchange processes, the Noah soil scheme (Chen and Dudhia, 2001) is utilized.

### 2.1 Framework

Figure 1 shows the framework of PAH extensions in WRF/Chem, where modules/subroutines that have been

modified in terms of embedding PAH extensions are listed. All the new variables related to emissions and concentration fields of PAHs and those intermediate variables used in different chemical and physical processes, such as air-soil gas exchange, gas phase/heterogeneous reactions, cloud scavenging, dry/wet deposition, adjective transport and cumulus convection are first defined in registry.chem and then included in respective sub-modules/routines.

The subroutine chem_driver is the main driver for handling chemistry related tasks on a particular time step, including emissions, photolysis, gas- and particulate-phase reactions, convective tracer transport, cloud chemistry, and dry/wet depositions. Based on the existing structure, all the chemical reactions involving gaseous and particulate PAHs mentioned in section 2.2.1 and 2.2.2 have been added to the RACMSORG_AQCHEM chemical mechanism by using KPP and the WRF/Chem KPP Coupler (Salzmann and Lawrence, 2006). Fixed rate coefficients are used for

gas-phase reactions, while for heterogeneous reactions oxidant/temperature/humidity -dependent functions are formulated. Gas-particle partitioning of semivolatile PAHs (and organic compounds in general) is implemented based on substance-specific empiric equilibrium relationships in addition to the MADE/SORGAM module. Dry deposition, wet deposition and wet scavenging from cumulus convection are calculated in the respective subroutines, using the dry deposition velocity and the fraction of gaseous species dissolved in cloud water calculated in



module_dep_simple and module_mosaic_wetscav, respectively. The air-soil gas exchange for organic compounds (secondary emissions) is implemented in addition to the primary emission module.

The simulated PAHs in our current WRF/Chem-PAH model include phenanthrene (PHE), chrysene (CHR) and BaP, representing volatile, semivolatile and non-volatile PAH compounds, respectively. The WRF/Chem-PAH framework
is not limited to these three species, and in general, all semivolatile compounds can be similarly implemented following our practices.

### 2.2 Atmospheric processes of PAHs

### 2.2.1 Gas-phase reactions

Reactions of gas-phase PAHs with hydroxyl radical (OH), $NO_3$ and $O_3$ are considered in this model. While $NO_3$
reactions appear to be less significant than OH reactions as the main PAH degradation process, the observed considerably high nitro-PAH yields suggest that night-time reactions of PAHs with $NO_3$ may be a significant contributor of these compounds in the atmosphere (Keyte et al., 2013). The significance of $NO_3$ oxidation is further discussed in Section 6. Here, the PAH oxidative loss is calculated as a second-order process using the model-predicted OH and $NO_3$ concentration (Table 1). At this stage, PAH reaction products are not tracked in the model.

### 2.2.2 Heterogeneous degradation of particulate BaP

In this study, we have developed and applied a more elaborate parameterization of the heterogeneous reaction kinetics of BaP degradation by ozone as a function of temperature and relative humidity based on the experimental data of Zhou et al. (2013) and the kinetic multi-layer model KM-SUB (Shiraiwa et al., 2010). Our model approach and parameterization build on a Langmuir-Hinshelwood reaction mechanism involving the decomposition of ozone
and formation of long-lived reactive oxygen intermediates (Shiraiwa et al., 2011; Berkemeier et al., 2016). Table S2 lists the parameterizations of the reaction rates. Text S1 reviews previous study of the heterogeneous degradation of BaP.

### 2.2.3 Gas/particle partitioning

The unmodified (conventional) configuration of WRF/Chem uses a thermodynamic equilibrium scheme (SORGAM)
to simulate gas-to-particle mass distribution of condensable and water-soluble species (Binkowski and Roselle, 2003). This approach is inappropriate for semivolatile and hydrophobic substances. Instead, empiric equilibrium partitioning expressions for PAHs are applied. We consider the absorption processes and chemically specific adsorption processes, both of which were found to be significant contributors to PAH gas-particle partitioning (Dachs and Eisenreich, 2000; Lohmann and Lammel, 2004).
We use a second equilibrium partitioning expression, which accounts for two contributions, absorption into organic matter and adsorption onto black carbon (BC) (Dachs and Eisenreich, 2000):

$$K_p = 10^{-12} \left( \frac{1.5 f_{oc}}{\rho_{oct} K_{OA}} + f_{bc} K_{SA} \right) = \frac{\sum c/c_{TSP}}{c_g} \qquad (2)$$



$$\theta = (1 + \frac{1}{K_p c_{TSP}})^{-1} \tag{3}$$

where $\rho_{oct}$ is the bulk density of octanol (0.82 kg L$^{-1}$), $f_{oc}$ is the organic carbon (OC) fraction of the particulate matter (the 1.5 multiplier converts OC to organic matter which is assumed to be well-represented by octanol), $K_{OA}$ is the octanol–air partition coefficient (dimensionless, temperature dependent; Odabasi et al. 2006), $f_{bc}$ is the BC

fraction of the particulate matter, $K_{SA}$ is the soot-air partition coefficient (L kg$^{-1}$), $\sum c_p$ is the particulate PAH concentration across all the size bins (ng m$^{-3}$), $c_{TSP}$ is the total particulate matter concentration (μg m$^{-3}$), and $c_g$ is the gas-phase concentration (ng m$^{-3}$).

Since direct $K_{SA}$ measurements are not available for PAHs, soot–air partitioning coefficients ($K_{SA}$ L kg$^{-1}$) are estimated as the ratios of soot–water ($K_{SW}$; Jonker and Koelmans, 2002) and the air–water ($K_{AW}$) partitioning

coefficients (dimensionless, temperature dependent; Bamford et al. 1999). Values of $K_{SW}$ vary substantially (up to a factor of 47 for the PAHs considered here) among relevant soot. The representative values for atmospheric BC are determined by weighting the reported $K_{SW}$ values by the contribution of their related combustion processes to the total emitted fine particulate matter used in the inventory of Galarneau et al. (2007).

### 2.2.4 Air-soil gas exchange

Semivolatile PAHs are subject to revolatilization (Lammel et al., 2009; Galarneau et al., 2014). An air-soil gas exchange module is therefore included in WRF/Chem-PAH. Air-soil gas exchange is parameterized following Strand and Hov (1996), which is based on Jury et al. (1983).

Model soil is a 0.15 m thick layer consisting of fixed volumes of soil organic matter, air and water. The soil layer is assumed to have the standard properties suggested by Jury et al. (1983) (Table 1). The change in PAHs

concentrations in soil/air, $c_s/c_a$, with time is expressed by:

$$\frac{\partial c_s}{\partial t} = \frac{1}{z_s}(F_{exc,soil} + F_{wet}) - k_{soil}c_s \tag{4}$$

$$\frac{\partial c_a}{\partial t} = -\frac{1}{z_a}F_{exc,soil} \tag{5}$$

where $z_s$ and $z_a$ are the soil and atmospheric layer depths (m), respectively, $F_{exc,soil}$ is the air-soil gas exchange flux, $F_{wet}$ is the wet deposition flux, and $k_{soil}$ is the degradation rate in soil. The air-soil gas exchange flux is given by:

$$F_{exc,soil} = v_s(c_a - \frac{c_s}{K_{soil-air}}) \tag{6}$$

where $v_s$ is the exchange velocity, $c_a$ is the PAHs concentrations in air, and $K_{soil-air}$ is the partitioning coefficient between soil and air. The exchange velocity is given by:

$$v_s = \frac{D_{air}a^{10/3}(1-l-a)^{-2} + D_{water}l^{10/3}K_{WA}(1-l-a)^{-2}}{z_s/2} \tag{7}$$

where $D_{air}$ is the air diffusion coefficient, $D_{water}$ is the liquid diffusion coefficient, $K_{WA}$ is the water-air partitioning

coefficient depending on the soil temperature and equals the inverse of $K_{AW}$, $l$ and $a$ are the water and air fractions in soil, respectively. Partitioning between soil and air is given by Karickhoff (1981):





$$K_{soil-air} = 4.11 \times 10^{-4} \times \rho_s f_{oc} K_{OA} \tag{8}$$

where $\rho_s$ is the soil density, $f_{oc}$ is the soil OC fraction and $4.11 \times 10^{-4}$ is a constant with units of $m^3\ kg^{-1}$. PAHs are subject to biodegradation in soil, processes which are actually not well quantified. The degradation rate in soil $k_{soil}$ is estimated to be $10^{-8}\ s^{-1}$ based on a laboratory model ecosystems study (Lu et al., 1977), following a global PAH
model (Friedman et al., 2014a; Friedman et al., 2014b).

In lack of monitoring data, the PAH concentrations in soil are initialized by the global multicompartmental model output: a pseudo-steady state of anthracene (ANT), fluoranthene (FLT) and BaP concentrations in the soil compartment had been safely reached by a global simulation over 10 years with 2.8° x 2.8° horizontal resolution (Lammel et al., 2009). PHE/CHR concentrations in soil are scaled from ANT/FLT according to the ratio upon
primary emission.  Figure S1 shows the air-soil gas exchange flux at a receptor site based on the above air-soil exchange scheme.

### 2.2.5 Wet and dry depositions

Dry deposition of gas phase species in WRF/Chem is treated using the standard resistance approach (Wesely, 1989). The original WRF/Chem routines have been adapted to include the deposition of gas-phase PAH compounds, and the
deposition flux is calculated from the product of the deposition velocity and gaseous PAH concentration in the lowermost model level. We consider the particulate phase PAH species to be bound to the atmospheric particulate matter in the accumulation mode, whose dry deposition flux is calculated using WRF/Chem particulate deposition parameterizations.

The model accounts for wet deposition of PAH species through the schemes for gas and particulate convective
transport, in-cloud and below-cloud scavenging of PAH species (sub-grid resolution, following UCI (University of California, Irvine) chemistry transport model; Neu and Prather, 2012).

### 3 Modification of transport scheme for low concentrated tracers

The transport of BaP seems stopped (Fig. 2a) when we follow WRF/Chem manual's suggestion by running with monotonic advection (chem, moist, scalar_adv_opt = 2) while the transport of other chemical tracers behaves
normally, e.g., BC, as shown in Fig. 2d. This is because the atmospheric concentration of BaP is too low, down to $10^{-9}$ to $10^{-12}$ ppmv. Other tracers are dealt with similarly, when artificially brought to extremely low concentrations, as confirmed for BC: The tracer does not undergo transport when dividing the BC concentration by $10^{10}$ before and then multiply by $10^{10}$ after the advection subroutine. Figure 2c and 2d shows the clear differences between the transportation of BC in these two cases. Besides, the near source concentration of BaP is too low compared with the
observation, because in the unmodified (conventional) transport non-zero BaP concentrations in air are limited to the immediate vicinities to strong sources and undergo fast degradation.

One of the important features of how the chemical transport model (Chem) couples with WRF in the WRF/Chem model is that the transport of chemical species is done by WRF. In WRF, monotonic advection is not a positive



definite option, so that mp_zero_out = 2 is usually set to make sure that the transport tendency of all the moisture variables will not grow below zero. To this end, the mp_zero_out_thresh is set as suggested to a small value of $10^{-8}$, and the transport tendency of moisture variables will be mapped to 0 when concentrations are smaller than mp_zero_out_thresh. In the coupled WRF/Chem model, when dealing with chemical transport, WRF advection

module treats all the chemical species as if they are moisture variables following the same criterion of exceedance of mp_zero_out_thresh. This is usually not a problem, because the concentrations of tracers transported are in general higher than this threshold. However, it is not the case for BaP, and the threshold truncates the BaP concentration and its transport tendency is forced to 0 and thus no transport occurs. To cope with this, we set mp_zero_out_thresh = $10^{-22}$ for PAHs species but leave it to $10^{-8}$ to moisture variables and all other chemical species. After this modification in

adjective transportation, BaP adequately undergoes transport in the model and the near source concentration of BaP is elevated too (Fig. 2b). This solution can be applied to all newly implemented low concentrated tracers in WRF/Chem.

## 4 Case study in East Asia

### 4.1 Model configuration

To apply the WRF/Chem-PAH model, we configured a domain that covers East China and Japan (15–55º N, 95–155º E) with a horizontal resolution of 27 km by 27 km and 39 vertical layers up to 0.01 hPa. The spatial coverage of the domain is shown in Fig. 2.

The physics options applied in this study are summarized as follows (also see Table S1): the Purdue–Lin scheme (Lin et al., 1983) is used for microphysics, which includes six classes of hydrometeors (water vapour, cloud water,

rain, cloud ice, snow and graupel). The planetary boundary layer is parameterized by the Mellor−Yamada−Janjic scheme (Janjic, 1994). It describes vertical sub-grid-scale fluxes due to eddy transport in the whole atmospheric column, while the horizontal eddy diffusivity is calculated with a Smagorinsky first-order closure. The surface layer parameterisation employed is the Eta Similarity surface layer scheme (Janjic, 1994). The land surface model to describe interactions between the soil and atmosphere is Noah Land Surface Model (Chen and Dudhia, 2001). The

Grell-3D Ensemble scheme (Grell and Devenyi, 2002) is used for cumulus parameterisation. The long- and shortwave radiation is calculated on-line with rapid radiative transfer model (Mlawer et al., 1997) and the Goddard schemes (Chou and Suarez, 1994), respectively. Photolysis rates are calculated using the Fast-J photolysis scheme (Wild et al., 2000) and updated every 60 minutes.

For simulation of standard aerosol precursors and aerosol species in the WRF/Chem model, anthropogenic emissions

for NOx, CO, non-methane volatile organic compounds, $SO_2$, $NH_3$, BC, and OC are taken from the EDGAR-HTAP global monthly inventory (http://edgar.jrc.ec.europa.eu/national_reported_data/htap.php) in the year 2010. The emissions of BC and OC in 2010 are further extrapolated to simulation year based on annual scaling factors taken from Lu et al. (2011), while no annual changes have been applied to emissions of other species. The EDGAR-HTAP inventory has a horizontal resolution of 0.1°. Hereby, biomass burning emissions are from the monthly Quick Fire



Emissions Dataset (QFED) (Darmenov and Silva, 2013). Biogenic volatile organic compounds emissions are calculated from the Model of Emissions of Gases and Aerosols from Nature (MEGAN) (Guenther et al., 2006). Anthropogenic PAH emissions are re-gridded from a $0.1° \times 0.1°$ global annual PAH emission inventory for the year 2008, with 69 detailed source types (Shen et al., 2013). For specific simulation period, monthly and annual scaling

factors in the simulated domain are taken from Zhang and Tao (2008) and Shen et al. (2013), respectively. A diurnal cycle of the PAH emissions are applied with two maxima, around 08:00 h and 19:00 h local time, following that of BC. Biogenic contributions to PAH emission have been neglected. Figure S2 shows the distributions of PHE, CHR and BaP emissions in the year 2008.

Meteorological initial and boundary conditions are based on the National Center for Environmental Prediction Final

Analysis' (NCEP-FNL) reanalysis data. Meteorology (temperature, horizontal wind, and moisture) nudged at all vertical levels. Chemical initial and boundary conditions of standard species are from the global Model for Ozone and Related Chemical Tracers (MOZART-4) (Emmons et al., 2010), simulations performed using $1.9° \times 2.5°$ horizontal resolutions. Initial PAH concentrations at all lateral boundaries are set to zero because China is the dominant emission country. To reach a steady state equilibrium concentration of PAHs in air, a spin-up time of 48

hours is used.

### 4.2 Model evaluation

The WRF/Chem-PAH model is developed to capture the PAH transport episode in higher temporal and spatial resolution, i.e. in diurnal to daily scales and in both concentration level and particulate mass fraction. To this end, two sets of continuous PAH field campaign data with at least daily resolution and in both gaseous and particulate

phases are chosen.

The first dataset provides both daytime and nighttime samples. As part of the Program of Campaigns of Air Quality Research in Beijing and Surrounding Region (CAREBeijing) 2013 campaign, measurement was made at the Xianghe Atmospheric Observatory (39.80° N, 116.96° E). The Xianghe site is located in a PAH near source area, 45 km southeast of Beijing and 70 km northwest of Tianjin (Fig. S2). The site is surrounded by residential suburban areas

and distanced some 5 km from the local town centre. Particulate- and gas-phase samples were collected twice a day (daytime samples 8:00 – 18:00 LT, nighttime samples 20:00 – 6:00 LT) during 11–22 July 2013. EC and OC samples of $PM_{10}$ were also collected separately twice a day (daytime samples: 6:00–18:00 LT, nighttime samples: 18:00–6:00 LT on the following morning) at the same site, which are important to evaluate gas/particle partition scheme of semivolatile PAHs. Details of the sampling methods and data quality control are described in Text S2.

Another PAH observation dataset is taken at the Gosan station (33.28° N, 126.17° E, 72 m above sea level) on Jeju Island, in the northern part of the East China Sea, about 100 km south of the Korean peninsula (Fig. S2). Gosan is a representative background site and an ideal location for studying long-range transport of air pollutants in East Asia (Han et al., 2006). Although the Gosan observation covers a long period from November 2001 to August 2003 (Kim et al., 2007; Kim et al., 2012), due to the high computational cost of WRF/Chem, we focus on an intensive

measurement period (14–25 February 2003) with continuous gas- and particulate phase PAHs (daily samples of



8:00–8:00 LT in the following morning) to represent a polluted continental outflow in East Asia in winter time. However, to further demonstrate the general model performance in a seasonal scale, we make additional simulation for a continuous summer period 6–17 June 2003. Details of sampling and analysis methods are given in Kim et al., 2012.

### 4.2.1 Evaluation at the near source areas

PAH diurnal variability are well captured for both gas- and particulate-phase species, with correlation coefficients r = 0.42–0.72 (Fig. 3). This demonstrates the model's good capability in predicting the vertical and horizontal transport of PAH. The simulated gas-phase PHE has the best correlation rate of 0.72 and also the best predicted average concentrations among the other PAH species: the observed (simulated) daily average concentrations of PHE gas are $25.6\pm13.8$ ($22.2\pm14.8$) ng m$^{-3}$, $13.6\pm5.1$ ($11.7\pm5.2$) ng m$^{-3}$ for daytime and $36.5\pm9.6$ ($32.6\pm13.8$) ng m$^{-3}$ for nighttime. The model catches the observed daily average concentration of particulate BaP (0.78 ng m$^{-3}$), and the night and day mean levels, 1.10 and 0.43 ng m$^{-3}$, respectively, are also predicted fairly well (1.37 and 0.14 ng m$^{-3}$, respectively). Both predicted gas- and particulate-phase CHR are overestimated (Fig. 3b–c). One reason may be that the same monthly and annual scaling factors of CHR emissions are applied all over the domain. Furthermore, as partitioning strongly influences atmospheric lifetime, the bias in predicted particulate mass fraction ($\theta$) can lead to the bias in predicted concentrations of CHR.

Although the simulated absolute values of $\theta$ are very close to the measured values (black solid and dotted lines in Fig. 4): the observed (simulated) average particulate mass fraction, is 0.52 (0.53) for 24 h, 0.49 (0.42) for daytime and 0.54 (0.65) for nighttime. The apparent good prediction might shield the fact of the overestimated OC and compensating underestimated BC concentrations (blue and red lines in Fig. 4). When applying the gas-particle partitioning parameterisations to the measured OC and BC concentrations, slightly lower than observed is predicted, but the correlation of simulated and observed $\theta$ is significantly improved, from r = 0.36 to r = 0.77 (green line in Fig. 4). The model's performance in simulating the changes of particulate mass fraction is quite satisfactory when the simulated bias in OC/BC concentration is eliminated. It also implies that adsorption to BC is more important than absorption by OC in determining partitioning, and that the partitioning scheme used in this model is suitable for this East Asian source area.

Overall, the model is found to predict the diurnal variations of PAH concentrations and particulate mass fractions reasonably well at the suburban site in the source region.

### 4.2.2 Evaluation of the Asian outflow

PAH predictions at remote sites are more challenging as the uncertainties in chemistry and gas-particle partitioning propagate. Model validation so far had been limited to seasonal features (Zhang et al., 2011a; Zhang et al., 2011b), while higher temporal features had not been addressed yet. For example, discrepancies of a factor of 16–476 between predicted and observed average PAH (BaP, CHR, BbF, BkF, IcdP, DahA, BghiP) concentrations at the Waliguan site, a continental background site for ambient air monitoring in western China, were found much larger than at



urban or suburban sites (Zhang et al., 2009). In our study, the predicted concentration levels in the Gosan winter case agree well with observations: the observed (simulated) average concentrations of PAHs are 0.020 (0.022) ng m$^{-3}$ for particulate BaP, 0.81 (1.73) ng m$^{-3}$ for gaseous PHE, 0.029 (0.029) ng m$^{-3}$ for gaseous CHR and 0.45 (0.24) ng m$^{-3}$ for particulate CHR (Fig. 5). Compared with previous studies, our simulated average concentrations of BaP agreed

very well with the observation, but Zhang et al. (2011a) underestimated BaP by about 50%. For the Gosan summer case, our simulated average BaP concentration is 0.006 ng m$^{-3}$ (Fig. S6), much closer to the observed 0.012 ng m$^{-3}$ than the simulated BaP concentration of ~0.001 ng m$^{-3}$ by Zhang et al. (2011a). In general, WRF/Chem-PAH model shows good/reasonable agreement with observations in both winter and summer seasons. However, it is worth notice that although the daily average concentration levels of PAHs are reasonably well simulated, the diurnal variations are

not well captured at the remote back ground site Gosan (Fig. S5).

The correlation of observed and simulated daily average CHR particulate mass fractions in the Gosan winter case is high (r = 0.73, see Fig. 6). The correlation is significantly lower at the Xianghe site (r = 0.37), which may due to the proximity to sources. The phase equilibrium of CHR may not be established shortly after emission and the model may not resolve its spatial gradients. An underestimation of simulated particulate mass fraction can be seen in the

both winter and summer cases (Fig. 6 and Fig. S6). Such underestimation may be caused by the combined effects of uncertainties, such as the emission, degradation, dry/wet deposition and the long-range transport of OC/BC.

These results suggest that our newly developed WRF/Chem-PAH is reasonably accurate in simulating the concentration levels and particulate mass fractions of PAHs for the Asian outflow.

### 4.3 Distributions of PAHs in East Asia

To illustrate the PAHs distribution in East Asia in both summer and winter time, Fig. 7 shows the surface concentrations of three representative PAHs averaged over summer 11–22 July 2013 and winter 14–25 February 2003 simulation periods. Another simulation period 6–17 June 2003 agrees with the period 11–22 July 2013 in reflecting summer time distribution characteristics. The lifetimes of PAH over Eastern China (20–42° N, 107–122° E mainland China) are 1.5–9 hours for PHE, 2–11 hours for CHR, 2 hours–3 days for BaP in summer and 9.5 hours–

3.5 days for PHE, 11 hours–4.5 days for CHR, 1.5–6.5 days for BaP in winter, respectively. Due to the relatively short lifetime of PAH species, the spatial distribution of PAH concentrations in the atmosphere is dominated largely by local emissions (Hafner et al., 2005) and the concentration of PAH decreases rapidly away from the source regions. There is a major eastward transport and outflow pathway that plumes with high levels of PAHs from the Eastern part of China are swept to the East China Sea and further to the western Pacific Ocean. The simulated

average concentration of 0.006 ng m-3 for particulate BaP during 14–25 February 2003 at a monitoring background site (26.19° N, 127.75° E) in Okinawa, Japan, which locates on the outflow pathway from China, is close to the monthly-averaged observation value 0.013 ng m$^{-3}$ (http://tenbou.nies.go.jp/gis/monitor/). The concentrations of PAHs are higher in winter than summer, mainly due to higher emissions and slower degradation rates.

The model calculated mean particulate mass fractions are also shown in Fig. 7. The particulate mass fractions are

higher in winter than in summer and in North China than in other regions. This is largely due to the distribution of



OC/BC concentrations (high in North China), and seasonal and latitudinal variation of temperature. PHE has extremely small particulate mass fractions, while θ BaP ≈ 1 over most of China in winter but only 95% over North China in summer. On the other hand, CHR shows the largest spatial and seasonal variations among these three compounds. Our predicted PAH distributions and particulate mass fractions generally agree with the previous studies in East Asia (Zhang et al., 2011b; Inomata et al., 2012).

## 5 Significance of PAH heterogeneous reaction with ozone and gas-phase reaction with nitrate radical

We test the impact of heterogeneous and homogeneous reaction of BaP, as well as $NO_3$ gas reactions of CHR and PHE. In other model studies of PAHs, these processes were usually neglected (Zhang et al., 2011a; Zhang et al., 2011b).

As shown in Fig. 8, the simulation without any BaP reaction significantly overestimates the average concentration of BaP. Compared with the kinetic model scheme, the model calculated BaP concentration increased from 0.14 to 1.32 ng m$^{-3}$ for daytime, 1.37 to 2.86 ng m$^{-3}$ for nighttime and 0.78 to 2.09 ng m$^{-3}$ for whole day, which move further away from the observed daily average of 0.78 ng m$^{-3}$ (Fig. 8). Furthermore, the homogeneous gas-phase reaction of BaP is of negligible efficiency as compared to the heterogeneous reaction. This confirms that BaP heterogeneous degradation is indispensable. A companion manuscript discussing the comparison of different heterogeneous degradation schemes is now in preparation.

Figure 9 shows the simulated concentrations of gas-phase PHE, gas- and particulate-phase CHR with and without $NO_3$ gas-phase reactions compared with observations at the Xianghe site. It is found that during nights with high $NO_3$ (48.5 and 43.0 ng m$^{-3}$, or ≈18 and ≈16 pptv as the 10 h mean, 12–14 July) the $NO_3$ reaction causes a significant night-time drop of PAH levels i.e., PHE and CHR by ≈-50% and -50 to -75%, respectively. This is surprisingly drastic for PHE regarding the rate coefficient, $k_{NO3} = 1.2\times10^{-13}$ cm$^3$ molec$^{-1}$ s$^{-1}$ (Table 1), corresponding to a lifetime of $\tau_{NO3} \approx 5$ h, but $\tau_{NO3} \approx 10$ min for CHR ($k_{NO3} = 4.0\times10^{-12}$ cm$^3$ molec$^{-1}$ s$^{-1}$, Table 1). The impact on the concentration of particulate-phase CHR is as significant as gas-phase CHR at night.

## 6 Conclusions and dicussion

We have developed the WRF/Chem-PAH model based on the WRF/Chem model to simulate the atmospheric fate of volatile, semivolatile and non-volatile PAH compounds. The implemented state-of-the-art processes for PAHs are: gas-particle partitioning, air-soil gas exchange, homogeneous gas-phase and heterogeneous reactions, cloud scavenging, dry and wet deposition, adjective transport and cumulus convection. The simulated PAHs in our current WRF/Chem-PAH model include PHE, CHR and BaP, representing volatile, semivolatile and non-volatile PAH compounds, respectively. Also, the model can be applied for any similar semivolatile trace organic compound.

The model has been applied for East Asia. The model predicts observations (both atmospheric concentration of PAHs and the particulate mass fraction of semivolatile CHR) at both a near-source and remote site in the continental





outflow reasonably well considering big uncertainties in our current knowledge, most notably with regard to the emission, gas-particle partitioning and atmospheric chemistry.

Both the testing of heterogeneous $O_3$ reaction and the homogeneous $NO_3$ reaction emphasise the importance of these reactions for the fate and distributions of the selected PAHs in the polluted atmospheric environment. PAH

modelling should include these reactions to better assess the fate and the distributions in polluted and remote environments. However, chemical kinetic measurement and understanding of pathways are limited, in particular for semivolatile species and for heterogeneous chemistry in general (Keyte et al., 2013). Intensive laboratory studies covering semivolatile PAHs, various aerosol matrices, and scenarios of particle mixing and aging are needed to improve PAH modelling.

The model accounts for secondary PAH emissions i.e., re-volatilisation from soil (semivolatile PAHs only). These emissions add to the pollutant distributions. However, as a consequence of prevailing westerly winds in combination with emissions being concentrated in Eastern China, unlike in South Asia (very little emissions in areas east of northeastern India and Bangla Desh) and other continents (Lammel et al., 2009; Galarneau et al., 2014), there is a large geographical overlap between secondary sources and primary sources over East Asia. Secondary emissions (re-

volatilisation) from the sea surface of the Yellow Sea and the adjacent shelf areas (East and South China Seas) also influence the regional PAH distributions over the mainland. This was not addressed in this study (process not included), and should be investigated, in order to better assess trans-Pacific transport of PAHs and their even more toxic metabolites, nitro-PAHs (Zhang et al., 2011a).

*Data availability*: The observation data and code of this study are available from the corresponding author upon reasonable request.

*Author contribution:* Y. F. Cheng, G. Lammel, U. Pöschl, and H. Su and conceived the study. Q. Mu did model development, case simulation, data processing and visualization. M. Shiraiwa, Y. F. Cheng, H. Su and U. Pöschl

developed the new heterogeneous degradation scheme for model implementation. C. N. Gencarelli and I. M. Hedgecock provided the code of WRF/Chem-Hg (Gencarelli et al., 2014) and helped with model development. Y. Chen contributed to data processing. P. Přibylová, M. Teich, Y. X. Zhang, G. J. Zheng, D. van Pinxteren, Q. Zhang and H. Herrmann provided the observation data at the Xianghe site. Y. F. Cheng, Q. Mu, G. Lammel, H. Su, U. Pöschl and P. Spichtinger discussed the results. Q. Mu, Y. F. Cheng G. Lammel and U. Pöschl wrote the manuscript.


*Competing interests:* The authors declare that they have no conflict of interest.

*Acknowledgements:* This work is supported by the Max Planck Society. The work of Y. F. Cheng and H. Su is also supported by the National Natural Science Foundation of China (41330635). We thank Young-Sung Ghim for

providing the Gosan measurement data. We thank Georg Grell and Bill Skamarock for the explanation of WRF/Chem transport scheme. The coding work is supported by Yvlu Qiu, Feng Wang, Mega Octaviani, Tabish



Ansari, Chao Wei and Stephan Nordmann. The description of the Xianghe sampling is supported by Pourya Shahpoury. We also thank Huizhong Shen, Rong Wang, Ye Huang, Fumo Yang and Pasquale Sellitto for valuable comments.

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



**Table 1. Physical and chemical properties of PAHs and soil.**

| Parameter | Symbol | Unit | PHE | CHR | BaP | Reference |
|---|---|---|---|---|---|---|
| molecular weight | $MW$ | g mol$^{-1}$ | 178.2 | 228.3 | 252.3 | |
| gas-phase OH reaction rate constant | $k_{OH}$ | cm$^3$ molec$^{-1}$ s$^{-1}$ | [a]3.1E-11 | [b]5.0E-11 | [b]1.5E-10 | [a]Atkinson et al. (1989); [b]Kloepffer and Wagner (2007) |
| gas-phase NO$_3$ reaction rate constant | $k_{NO3}$ | cm$^3$ molec$^{-1}$ s$^{-1}$ | [a]1.2E-13 | 4E-12 | [b]5.4E-11 | [a]Kwok et al. (1994); [b]Kloepffer and Wagner (2007) |
| gas-phase O$_3$ reaction rate constant | $k_{O3}$ | cm$^3$ molec$^{-1}$ s$^{-1}$ | [a]4.0E-19 | [b]4.0E-19 | [b]2.6E-17 | [a]Kwok et al. (1994); [b]Kloepffer and Wagner (2007) |
| soot-water partitioning coefficient | $K_{sw}$ | L kg$^{-1}$ | 4.34E+05 | 2.82E+07 | 9.59E+07 | Jonker and Koelmans (2002) |
| octanol-air partitioning coefficient | $K_{OA}$-m | dimensionless | 3293 | 4754 | 5382 | Odabasi et al. (2006) |
| | $K_{OA}$-b | dimensionless | -3.37 | -5.65 | -6.5 | log $K_{OA}$ = m/T(K) + b |
| air-water partitioning coefficient | $K_{AW}$-m | dimensionless | -5689.2 | -12136.16 | -4437.1 | Bamford et al. (1999) |
| | $K_{AW}$-b | dimensionless | 12.75 | 32.235 | 3.9881 | ln $K_{AW}$ = m/T(K) + b |
| soil degradation rate | $k_{soil}$ | s$^{-1}$ | 1.00E-08 | 1.00E-08 | 1.00E-08 | Mackay and Paterson (1991) |
| water content of soil | $l$ | dimensionless | 0.3 | 0.3 | 0.3 | Jury et al. (1983) |
| air content of soil | $a$ | dimensionless | 0.2 | 0.2 | 0.2 | Jury et al. (1983) |
| soil depth | $z_s$ | m | 0.15 | 0.15 | 0.15 | Jury et al. (1983) |
| bulk density of soil | $\rho_s$ | kg m$^{-3}$ | 1350 | 1350 | 1350 | Jury et al. (1983) |
| organic carbon fraction of soil | $f_{OC}$ | dimensionless | 0.0125 | 0.0125 | 0.0125 | Jury et al. (1983) |
| air diffusion coefficient of soil | $D_{air}$ | m$^2$ s$^{-1}$ | 5.00E-06 | 5.00E-06 | 5.00E-06 | Jury et al. (1983) |
| water diffusion coefficient of soil | $D_{water}$ | m$^2$ s$^{-1}$ | 5.00E-10 | 5.00E-10 | 5.00E-10 | Jury et al. (1983) |





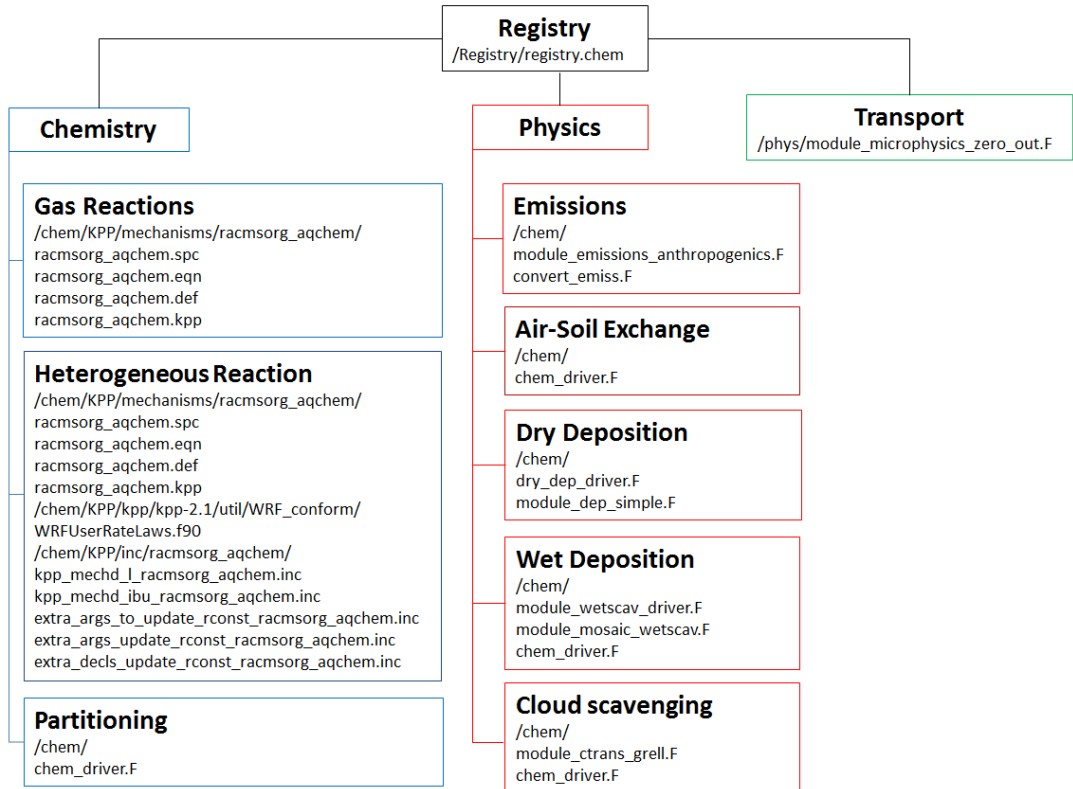

**Figure 1. Framework of PAH extensions in WRF/Chem.**



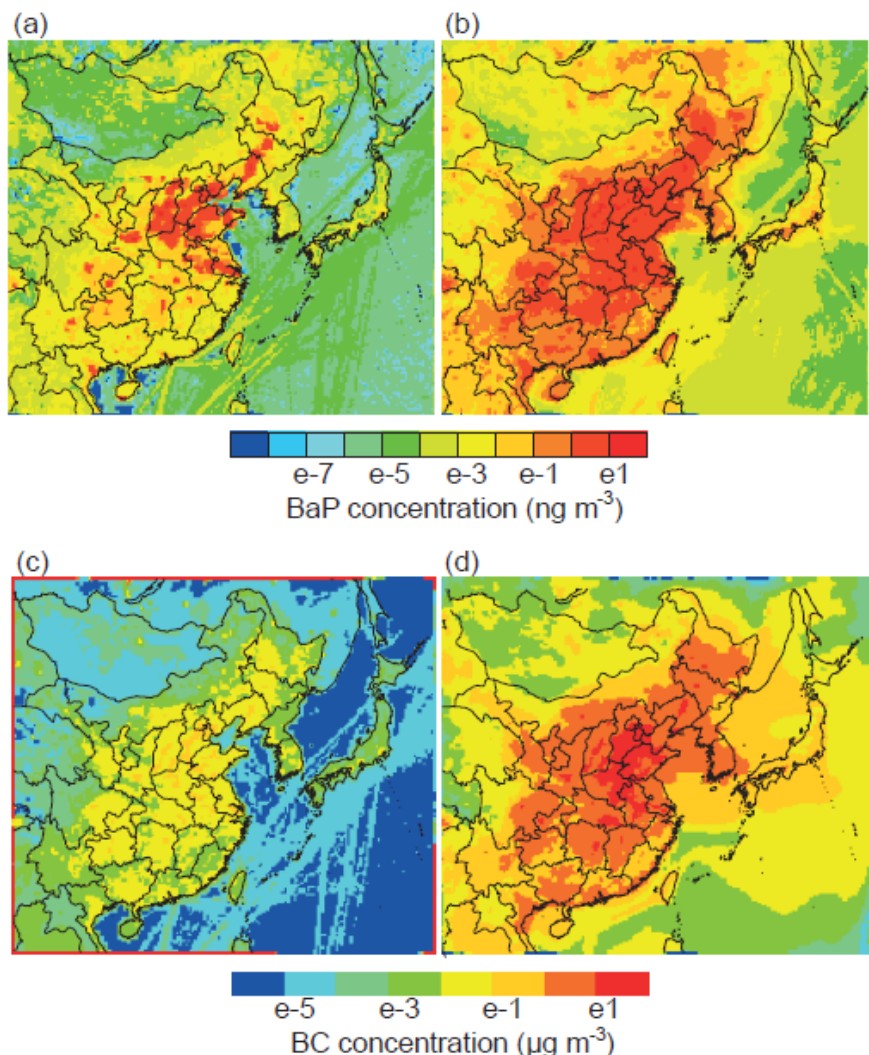

Figure 2. Simulated near-ground concentrations of BaP with (a) conventional transport scheme and (b) modified transport
       scheme for low concentrated species, and BC with (c) scaled low concentration and (d) normal concentration with
       conventional transport scheme averaged on 14 February 2003.





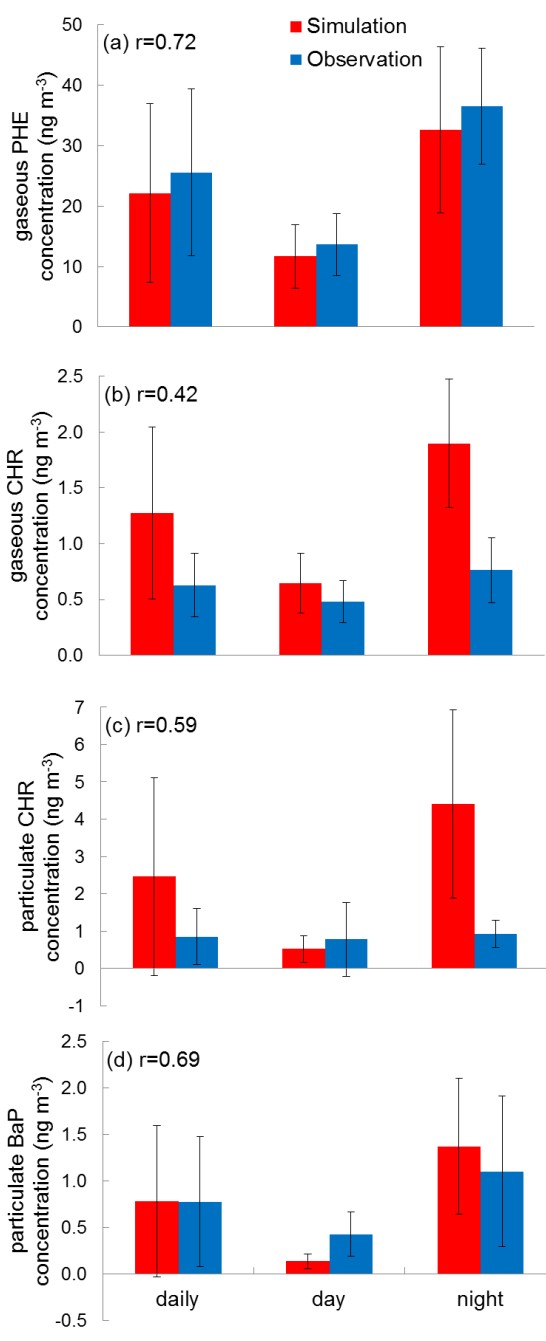

**Figure 3. Simulated and observed concentrations of (a) gaseous PHE, (b) gaseous CHR, (c) particulate CHR and (d) particulate BaP at the Xianghe site averaged over 11–22 July, 2013. Error bars show the standard deviations. The correlation coefficient is indicated by r.**




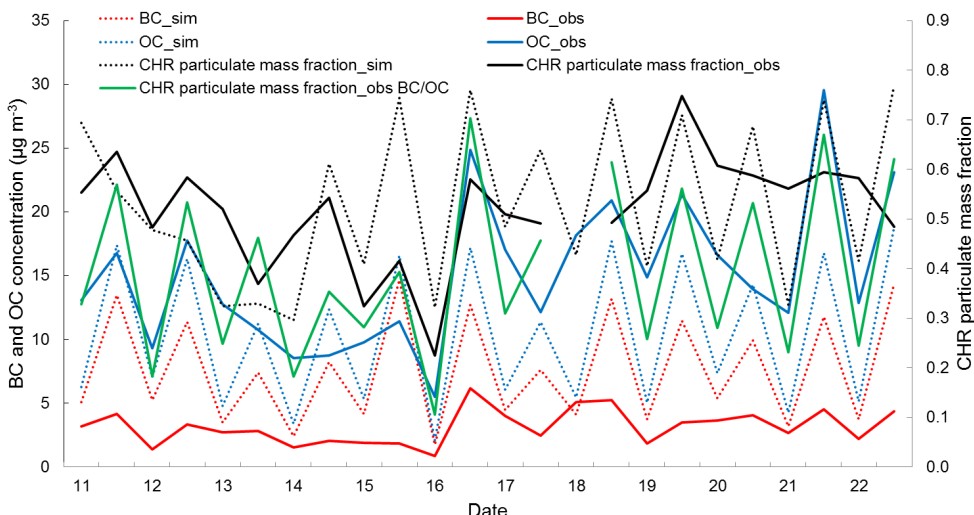

**Figure 4. Simulated and observed BC, OC and particulate mass fraction of CHR at the Xianghe site during 11–22 July, 2013 (10 h means). Calculated particulate mass fraction of CHR based on observed BC and OC is shown in green solid line. Calculation explained in the text.**



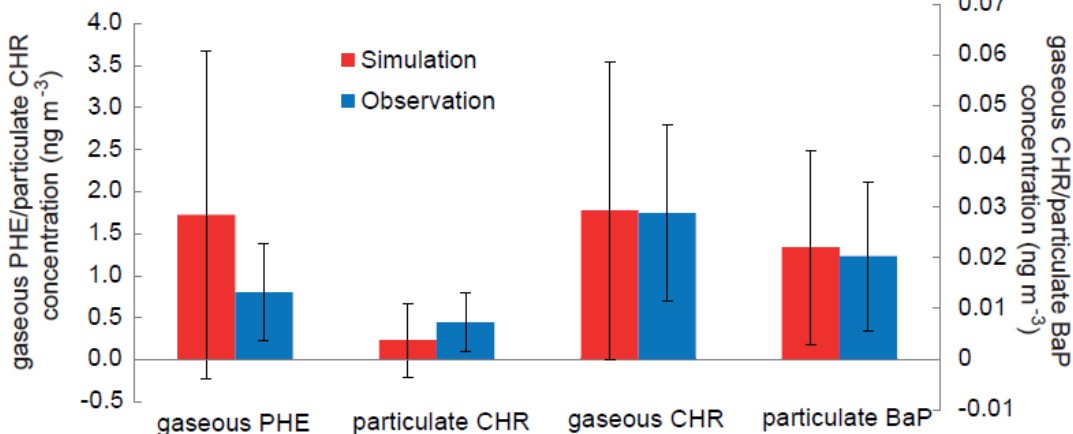

**Figure 5. Simulated and observed concentrations of gaseous PHE, particulate CHR, gaseous CHR and particulate BaP at the Gosan site averaged over 14–25 February 2003. Error bars show the standard deviations.**



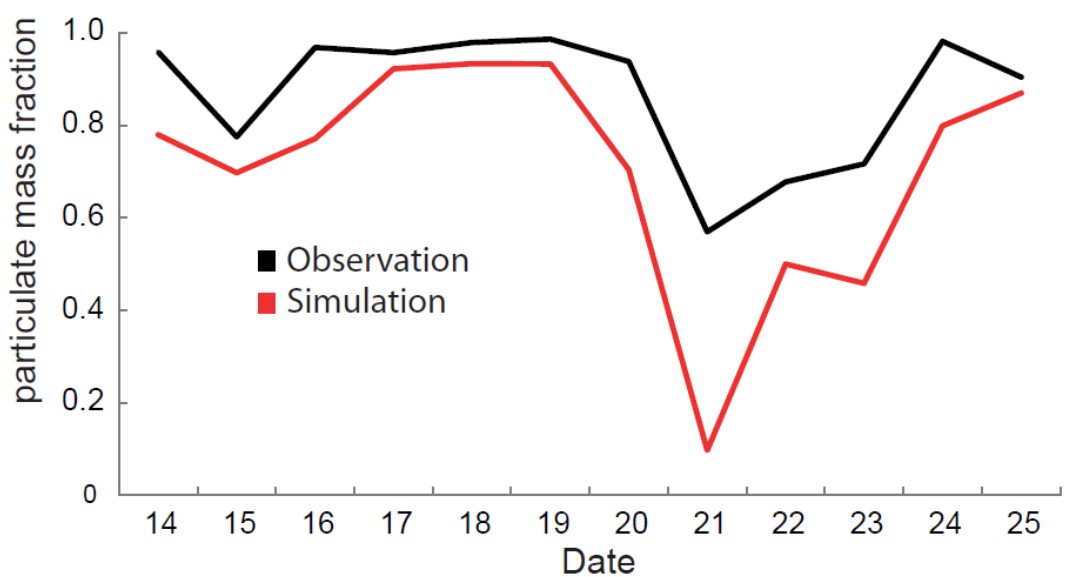

**Figure 6. Simulated and observed daily averaged particulate mass fraction of CHR at the Gosan site during 14–25 February, 2003.**




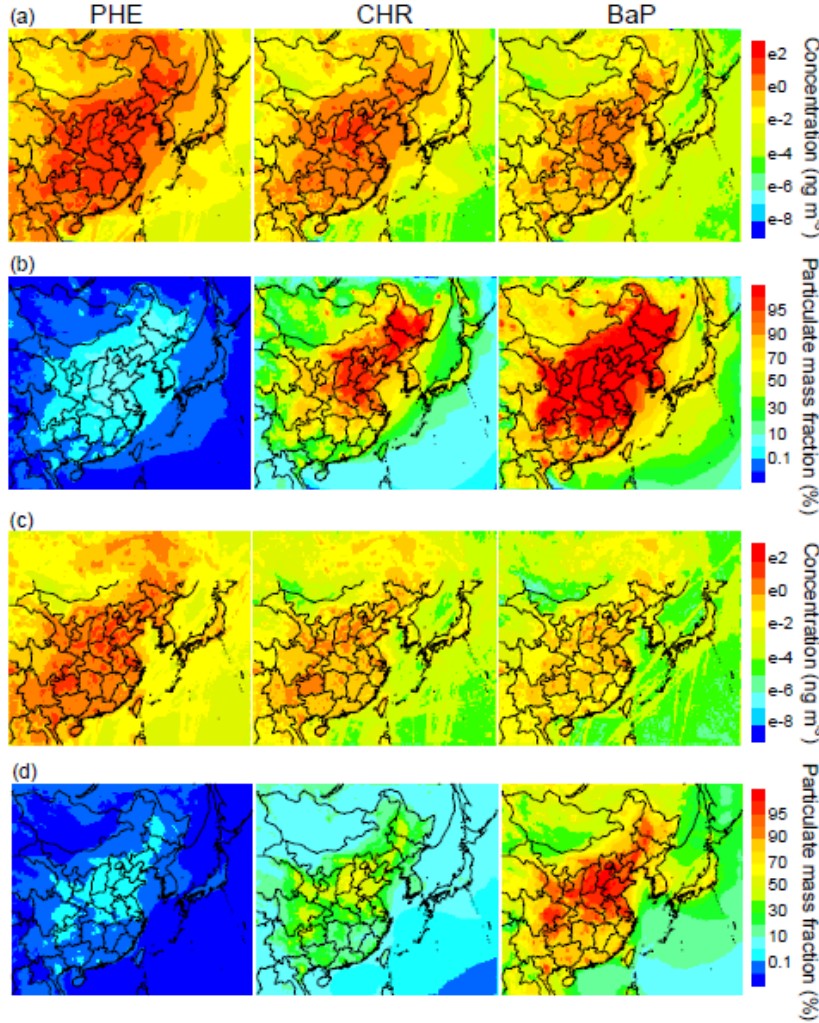

**Figure 7. Simulated (a) surface concentrations and (b) particulate mass fractions of PHE, CHR and BaP averaged over 14–25 February, 2003. Simulated (c) surface concentrations and (d) particulate mass fractions of PHE, CHR and BaP averaged over 11–22 July, 2013.**





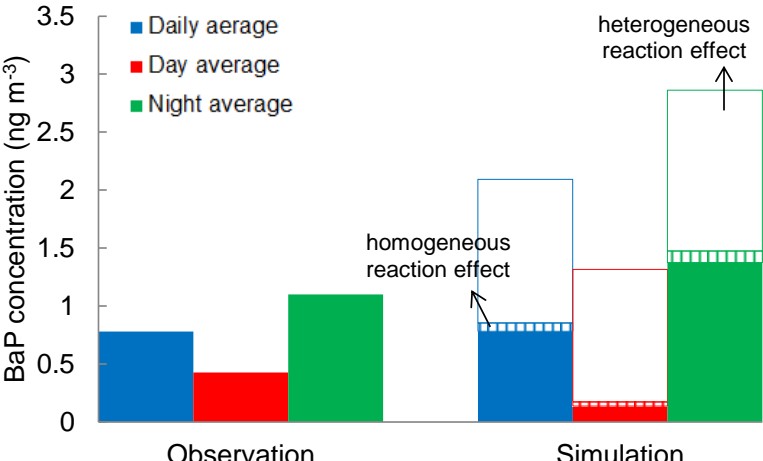


**Figure 8. Simulated concentrations of BaP compared with observation at the Xianghe site during 11–22 July, 2013. The contributions of heterogeneous and homogeneous reaction are shown in blank and vertical bar, respectively.**






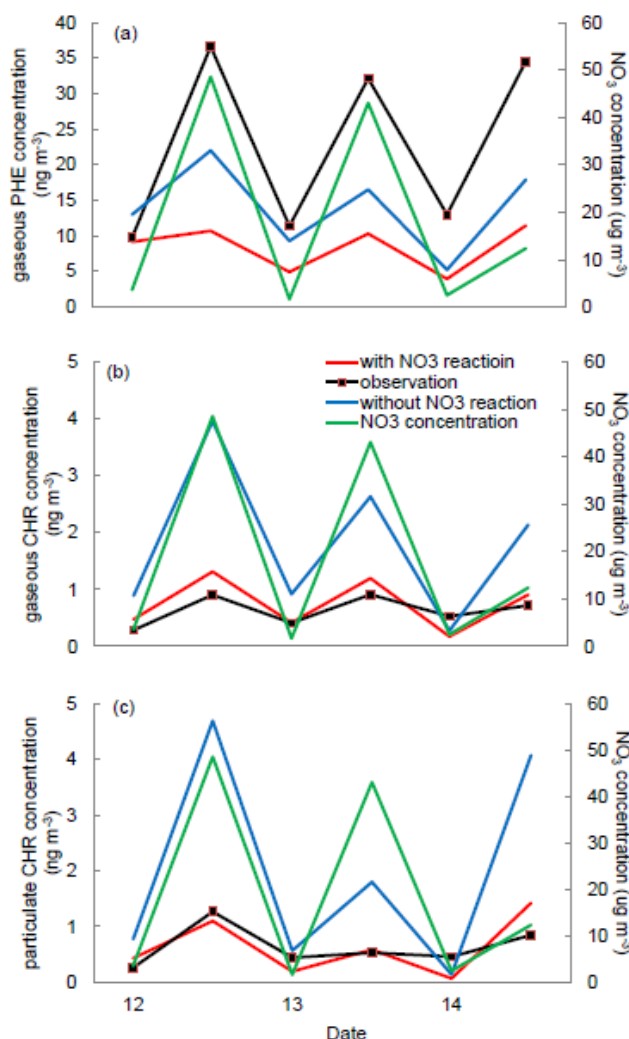

**Figure 9. Simulated concentrations of (a) gaseous PHE, (b) gaseous CHR and (c) particulate CHR with and without reactions of NO₃ compared with observation and simulated concentrations of NO₃ during 12–14 July 2013.**