# Peer review of ""Regional modelling of polycyclic aromatic hydrocarbons: WRF/Chem-PAH model development and East Asia case studies""

_Atmospheric Chemistry and Physics, 2017_

## Referee Comment (RC1) · Anonymous Referee #1 · 19 Jul 2017

Thank you for your revisions, they were helpful in answering some of the questions I had in the previous version. I was disappointed that a table of results was not provided.

I am still unconvinced by the model performance data. For example, in Fig.3, the authors present 11 days of data. These can be examined full day, daytime only, and nighttime only. The only metric produced to compare these two data sets is a correlation coefficient. It's not clear which two sets of data the correlation is between and it's not clear to me that a simple correlation is meaningful here. For Fig3c, the correlation for particulate chrysene is provided as 0.59, but the night time simulation looks completely uncorrelated in the graph. The only graph where all three - full day, day,

night - look reasonably similar are the gas-phase phenanthrene comparisons. Some of the poor performance may be due to using summer samples, when chrysene and benzo[a]pyrene concentrations are relatively low. More clarification is definitely needed and perhaps a metric that takes into account absolute differences as well.

---

## Referee Comment (RC2) · Anonymous Referee #2 · 28 Jul 2017

PAHs are substances the can severely impair human health and ecosystems. However, still little is known about their fate in the environment, in particular in the atmospheric environment. By investigating the atmospheric transport of some important PAHs with the state-of-the art model WRF/Chem in China, a region where the pollution by PAHs is supposed to be significant, the authors provide an important contribution to PAH research. The physical-chemical processes they implement into WRF/CHEM are based on sound science. Also, the general setup of the model study appears sound to me.

However, I'm not convinced that the authors treat the PAH emissions - a crucial part of modeling studies - in an appropriate way. They use emissions of 2008 and compare

model results to measurements of 2013 and 2003. This only makes sense if PAH emissions did not change within this period. On the other hand, the authors convincingly explain the increasing relevance of PAHs in Asia due to rapidly increasing emissions. If the authors applied inter-annual scaling factors for modeling the years where the measurements were carried out, they should explain in detail how this was done. Otherwise, they should comment on this contradiction. An agreement with measurements alone could also be "right for the wrong reason".

For evaluating the model against measurements the authors compare arithmetic means and Pearson's correlation coefficients of time series, I assume (they don't mention it explicitly). In figure 3 and 5 one can see that the error bars reach negative values. This indicates non-normal distributions and statistical measures like arithmetic mean and standard deviations cannot be applied. The authors must check the distribution and decide based on this which measures to use.

The authors use expressions like "good prediction", "fair agreement", "significantly improved", ... to judge their model results. They should explain by which criteria they consider a result (an average or a correlation) as good or not good. As they didn't perform any statistical tests it seems to be pure opinion. I found only one statement where they explain their opinion: Compared with previous studies ... (page 10, line 4).

Minor comments:

page 1 line 29: To my knowledge the word "tracer" is used for inert substances (which PAHs are not). For substances in very low concentrations I would rather suggest to use "trace gases/compounds/substances".

page 2 line 27ff: the information of item 4 is included in item 1 and could be left away.

Figures 2c and 2d are not necessary because the authors explain in the text (page 6 line 32ff) why the transport behavior of trace substances is inherent to the model.

---

## Author Comment (AC1) · 23 Aug 2017

**Response to comments of referee #1**

**General comment:**

Thank you for your revisions, they were helpful in answering some of the questions I had in the previous version. I was disappointed that a table of results was not provided.

**Response:**

Thanks for the constructive comments on both our current and previous versions, which help us to improve our manuscript.

Upon requests, we now add the Table R1 (SI Table S3) and Table R2 (SI Table S4) to show detailed statistic metrics about the Xianghe summer case and the Gosan winter case, respectively. Please kindly find more details in our point-to-point reply to your specific comments.

Also, in this response and the revised manuscript,

(1) We summarize the motivation of model development and new features in the newly developed WRF-Chem-PAH mode.

(2) Regarding model evaluation, we clarify the selection of evaluation data sets in detail. For model performance on seasonal bases, we added Gosan summer case and compared the model performance with previous global models. To further demonstrate that our model performs at least similarly with previous regional models in East Asia, we choose the same simulation period in Beijing as by Inomata et al. (2012) and compare the daily concentrations of the same species (total CHR and total BaP) with that study. The comparisons show that the two models perform quite similar and more details will be presented in the following response.

**Specific comments:**

I am still unconvinced by the model performance data. For example, in Fig.3, the authors present 11 days of data. These can be examined full day, daytime only, and nighttime only. The only metric produced to compare these two data sets is a correlation coefficient. It's not clear which two sets of data the correlation is between and it's not clear to me that a simple correlation is meaningful here. For Fig3c, the correlation for particulate chrysene is provided as 0.59, but the night time simulation looks completely uncorrelated in the graph. The only graph where all three - full day, day, night - look reasonably similar are the gas-phase phenanthrene comparisons. Some of the poor performance may be due to using summer samples, when chrysene and benzo[a]pyrene concentrations are relatively low. More clarification is definitely needed and perhaps a metric that takes into account absolute differences as well.

**Response:**

The correlation coefficients shown in Fig. 3 use combined daytime and nighttime data sets, i.e., between 24 observation and 24 simulation samples (12 daytime and 12 nighttime). These correlation coefficients have all passed the Student's t-test with a significance level of 0.05. Correlation coefficients for only daytime or nighttime samples are not included, because of small data sets. In Fig. 5, correlation coefficients are also not shown for the same reason. We have added the clarifications of correlation coefficients to the caption of Fig. 3: *The correlation coefficients use combined daytime and nighttime data sets, passing the Student's t-test with a significance level of 0.05.*

We add the Table R1 (SI Table S3) and Table R2 (SI Table S4) to show detailed statistic metrics about the Xianghe summer case and the Gosan winter case, respectively. Table R1 (SI Table S3) shows that our model performance in Xianghe summer is as good as the previous modeling study in Beijing (Xianghe is a semi-urban town in the Beijing metropolitan area) comparing with daily PAH observation (Inomata et al., 2012). Their simulated daily average/median concentrations of PAH are 0.1–2 factors of observation, while in our summer case these are a factor of 0.7–3; their correlation coefficients between simulation and observation are 0.30–0.58, while in our summer case these are 0.42–0.72. Our diurnal comparisons reveal that the overestimate of daily CHR concentrations mainly comes from nighttime rather than daytime, both for gaseous and particulate CHR. The following discussion was added: Page 9 line 8, "*PAH diurnal variabilities are well captured for both gas- and particulate-phase species at the Xianghe site, with correlation coefficients of 0.42–0.72 (Fig. 3, Table S3) compared with 0.30–0.58 in Beijing (Xianghe is a semi-urban town in the Beijing metropolitan area) by Inomata et al. (2012)*". Page 9 line 14, "*The model well catches the observed daily average concentration of particulate BaP (observation 0.78 ng m$^{-3}$, simulation 0.78 ng m$^{-3}$), while Inomata et al. (2012) underestimated daily concentration of BaP in Beijing by about a factor of 2. Page 9 line 18, "Further diurnal comparisons reveal that such overestimate of daily CHR concentrations mainly comes from nighttime rather than daytime (Table S3).*"

**Motivation and new feature of the model:**

The WRF/Chem-PAH model has been developed to resolve detailed transport and transformation processes of PAHs, particularly in high temporal and spatial resolution. Regarding the model development, our WRF/Chem-PAH model reflects the state-of-the-art and the up-to-date understanding of current PAHs studies with several new or updated features. (1) The gas-phase reactions include not only reactions with OH but also reactions with previous neglected $O_3$ and $NO_3$ radical. We found that during nights with high $NO_3$, the $NO_3$ reaction causes a significant night-time drop of gaseous PAH levels by about -50 – -75%. (2) The heterogeneous degradation of particulate BaP is treated with a new elaborated kinetic scheme, which considers

aerosol phase change and chemical reactivity under different temperature and humidity. Compared with previous BaP degradation schemes, the new scheme greatly improves model performance in both near source summer time (e.g., Xianghe site) and remote winter time (e.g., Gosan site) cases (on-going project). (3) The gas/particle partitioning considers absorption into organic matter and adsorption onto soot, which is an improvement from the commonly used adsorption to unspecific aerosol surfaces i.e., Junge-Pankow scheme. (4) The air-soil gas exchange process embedded in the WRF/Chem-PAH module is important for semi-volatile PAHs under certain atmospheric conditions (another on-going project), but neglected by some previous models. (5) We have successfully demonstrated an example to implement low concentrated tracers in WRF/Chem (Section 2.1) and solved the crucial transport problem of low concentrated tracers (Section 3). The method and solution can be adapted to other atmospheric low concentrated species by WRF/Chem users.

**Model evaluation:**

Previous global or regional models have demonstrated their ability to reproduce seasonal or annual variations of PAHs in the atmosphere, but to our knowledge, none of them compared with observation data on a diurnal basis, let alone in both atmospheric phases.

We use the Xianghe summer data since it is the only available observation in East Asia that provides continuous measurement both in daytime and nighttime and in both gaseous and particulate phases. Previous modeling studies (e.g. Zhang et al., 2011a; Zhang et al., 2011b; Inomata et al., 2012; Inomata et al., 2013; Sehili and Lammel, 2007; Lammel et al., 2009; Friedman and Selin, 2012; Friedman et al., 2014; Shen et al., 2014) were not able to do so. Simultaneous EC and OC data are also available for evaluation of gas/particle partitioning. Moreover, since East Asia in summer often experiences complex atmospheric conditions (e.g. East Asia summer monsoon), the model's good performance in Xianghe summer case implies its ability in representing complex atmospheric conditions.

The Gosan winter data is used to validate model performance in winter. An important reason to choose a winter episode at the Gosan site is that the continental outflow and emissions are both strongest in this season. This winter episode is designated as "continental outflow conditions" and "pollution period" which transports PAH from continental sources to Gosan (Kim et al., 2007; Kim et al., 2012).

However, to evaluation model performance in a seasonal basis, we do have included a simulation of a summer episode at the Gosan site (Fig. S7, continuous summer period 6–17 June 2003 when all the simulated species are available). In winter, our simulated average BaP concentration is $\approx 0.022$ ng m$^{-3}$ at Gosan site, in good agreement with the observed 0.020 ng m$^{-3}$, while the simulated BaP was underestimated by about 50% in Zhang et al. (2011a). For the summer case, our simulated average BaP concentration is $\approx 0.006$ ng m$^{-3}$, much closer to the

observed value of 0.012 ng m$^{-3}$ (Fig. S7) than the previously simulated BaP concentration of 0.001 ng m$^{-3}$ in Zhang et al. (2011a). No simulated values of other species have been reported by Zhang et al. (2011a). In general, WRF/Chem-PAH model shows good/reasonable agreement with observations in both winter and summer seasons.

The simulation periods are short in Gosan because only few periods could be covered from November 2001 to August 2003 with the longest consecutive measurements in each season not exceeding 15 days (Kim et al., 2012). Only total PAHs without consideration of single species have been used for model evaluation in several previous studies (Zhang et al., 2009; Zhang et al., 2011a; Zhang et al., 2011b). In Kim et al. (2012), even larger data gaps apply for monitoring of individual PAH species at Gosan: the spring episode (28 March – 11 April 2002) missed 6 out of total 15 samples of gaseous PHE, the summer episode (18 August – 1 September 2003) missed all data of particulate CHR and the fall episode (12–26 November 2001) missed all data of gaseous BaP. Only the winter episode (14–25 February 2003) has the complete daily observation of the simulated PAH species in both particulate and gaseous phases. Data availability is upon personal communication with Prof. Young-Sung Ghim.

In general, the observation data set is much smaller in Asia, unlike in North America or Europe where continuous monitoring of PAH at many background sites are available. There is no PAH monitoring in China, while the one in Korea (only one Gosan site) and Japan are discontinuous (e.g. 1 day/month in Japan) and/or cover only particulate phase. Campaigns are also limited. For example in Inomata et al. (2012), the simulated concentrations of nine particulate PAHs agreed very well with the measured concentrations, but it is a pity that there were only particulate PAH observation available and the Noto site was monitored at 1-week intervals. Hence, more model process evaluation in East Asia on the seasonal scale is only possible when more observation data is available in the future.

To further demonstrate that our model performs at least similarly with previous regional models in East Asia, we choose the same simulation period in Beijing as by Inomata et al. (2012) and compare the daily concentrations of the same species (total CHR and total BaP) with that study. Figure R1 shows that the RAQM2-POP model underestimated average CHR by about 15% while our WRF/Chem-PAH model overestimated by 3%; the RAQM2-POP model overestimated average BaP by about 50% while our WRF/Chem-PAH model underestimated by about 50%. It is worth noticing that, without knowing the exact starting and ending hours of the daily samples, we arbitrarily choose 08:00 (local hour) to the next day 08:00 as our simulated daily averages, which may lead to a slight time shift in the comparison with observations. Apparently, the two models perform quite similar.

Table S1 (SI Table S3). Observed (obs) and simulated (sim) mean concentration, median, standard deviation ($\sigma$), mean bias (MB), root mean square error (RMSE), mean absolute deviation (MAD) in unit ng m$^{-3}$ and correlation coefficient (R) at the Xianghe site averaged over 11–22 July, 2013. The correlation coefficients use combined daytime and nighttime data sets, passing the Student's t-test with a significance level of 0.05.

**gaseous PHE**

| | daily | | | | day | | | | night | | | |
| --- | --- | --- | --- | --- | --- | --- | --- | --- | --- | --- | --- | --- |
| | obs | sim | sim-obs | (sim-obs)/obs | obs | sim | sim-obs | (sim-obs)/obs | obs | sim | sim-obs | (sim-obs)/obs |
| Mean | 25.6 | 22.2 | -3.4 | -13.4% | 13.6 | 11.7 | -1.9 | -14.1% | 36.5 | 32.6 | -3.9 | -10.7% |
| Median | 23.9 | 15.5 | -8.5 | -35.5% | 13.0 | 12.5 | -0.5 | -4.2% | 33.3 | 35.3 | 2.0 | 6.0% |
| $\sigma$ | 13.8 | 14.8 | 0.9 | 6.7% | 5.1 | 5.2 | 0.1 | 2.5% | 9.6 | 13.8 | 4.2 | 43.8% |
| MB | -3.0 | | | | -2.0 | | | | -3.9 | | | |
| RMSE | 10.5 | | | | 5.4 | | | | 13.6 | | | |
| MAD | 7.9 | | | | 4.5 | | | | 11.1 | | | |
| R | 0.72 | | | | | | | | | | | |

**gaseous CHR**

| | daily | | | | day | | | | night | | | |
| --- | --- | --- | --- | --- | --- | --- | --- | --- | --- | --- | --- | --- |
| | obs | sim | sim-obs | (sim-obs)/obs | obs | sim | sim-obs | (sim-obs)/obs | obs | sim | sim-obs | (sim-obs)/obs |
| Mean | 0.63 | 1.27 | 0.65 | 102.8% | 0.48 | 0.65 | 0.17 | 34.5% | 0.76 | 1.90 | 1.14 | 149.2% |
| Median | 0.56 | 1.08 | 0.52 | 94.0% | 0.42 | 0.61 | 0.19 | 45.5% | 0.81 | 1.98 | 1.17 | 143.8% |
| $\sigma$ | 0.28 | 0.77 | 0.49 | 170.9% | 0.19 | 0.27 | 0.08 | 42.6% | 0.29 | 0.57 | 0.28 | 97.5% |
| MB | 0.67 | | | | 0.16 | | | | 1.14 | | | |
| RMSE | 0.97 | | | | 0.35 | | | | 1.31 | | | |
| MAD | 0.74 | | | | 0.30 | | | | 1.14 | | | |
| R | 0.42 | | | | | | | | | | | |

**particulate CHR**

| | daily | | | | day | | | | night | | | |
| --- | --- | --- | --- | --- | --- | --- | --- | --- | --- | --- | --- | --- |
| | obs | sim | sim-obs | (sim-obs)/obs | obs | sim | sim-obs | (sim-obs)/obs | obs | sim | sim-obs | (sim-obs)/obs |
| Mean | 0.85 | 2.46 | 1.61 | 189.6% | 0.78 | 0.52 | -0.26 | -33.4% | 0.92 | 4.40 | 3.48 | 378.0% |
| Median | 0.58 | 0.98 | 0.40 | 69.1% | 0.45 | 0.45 | 0.00 | 0.7% | 0.97 | 4.89 | 3.92 | 404.4% |

| | obs | sim | sim-obs | (sim-obs)/obs | obs | sim | sim-obs | (sim-obs)/obs | obs | sim | sim-obs | (sim-obs)/obs |
|---|---|---|---|---|---|---|---|---|---|---|---|---|
| σ | 0.75 | 2.65 | 1.90 | 252.7% | 0.99 | 0.35 | -0.64 | -64.7% | 0.37 | 2.52 | 2.15 | 579.3% |
| MB | | 1.83 | | | | 0.03 | | | | 3.48 | | |
| RMSE | | 3.07 | | | | 0.46 | | | | 4.23 | | |
| MAD | | 2.00 | | | | 0.35 | | | | 3.51 | | |
| R | | 0.59 | | | | | | | | | | |

**particulate BaP**

| | daily | | | | day | | | | night | | | |
|---|---|---|---|---|---|---|---|---|---|---|---|---|
| | obs | sim | sim-obs | (sim-obs)/obs | obs | sim | sim-obs | (sim-obs)/obs | obs | sim | sim-obs | (sim-obs)/obs |
| Mean | 0.78 | 0.78 | 0.00 | 0.3% | 0.43 | 0.14 | -0.29 | -68.1% | 1.10 | 1.37 | 0.27 | 24.5% |
| Median | 0.49 | 0.29 | -0.20 | -41.1% | 0.39 | 0.13 | -0.26 | -67.1% | 1.50 | 0.89 | -0.61 | -40.6% |
| σ | 0.81 | 0.70 | -0.12 | -14.5% | 0.24 | 0.08 | -0.16 | -67.5% | 0.81 | 0.73 | -0.08 | -10.1% |
| MB | | 0.002 | | | | -0.29 | | | | 0.27 | | |
| RMSE | | 0.61 | | | | 0.38 | | | | 0.76 | | |
| NMB | | 0.48 | | | | 0.31 | | | | 0.63 | | |
| R | | 0.69 | | | | | | | | | | |

Table R2 (SI Table S4). Same as SI Table S3 but at the Gosan site averaged over 14–25 February, 2003.

| | obs | sim | sim-obs | (sim-obs)/obs |
|---|---|---|---|---|
| **gaseous PHE** | | | | |
| Mean | 0.81 | 1.73 | 0.92 | 113.6% |
| Median | 0.54 | 1.14 | 0.60 | 109.8% |
| σ | 0.57 | 1.95 | 1.38 | 242.1% |
| MB | 0.92 | | | |
| RMSE | 2.35 | | | |
| MAD | 1.32 | | | |
| **gaseous CHR** | | | | |
| Mean | 0.03 | 0.03 | 0.000520 | 1.8% |
| Median | 0.02 | 0.02 | 0.00 | -4.6% |
| σ | 0.02 | 0.03 | 0.01 | 69.8% |
| MB | 0.00 | | | |
| RMSE | 0.04 | | | |
| MAD | 0.03 | | | |
| **particulate CHR** | | | | |
| Mean | 0.45 | 0.24 | -0.21 | -47.5% |
| Median | 0.40 | 0.06 | -0.34 | -84.1% |
| σ | 0.35 | 0.44 | 0.09 | 26.2% |
| MB | -0.21 | | | |
| RMSE | 0.51 | | | |
| MAD | 0.36 | | | |
| **particulate BaP** | | | | |
| Mean | 0.020 | 0.022 | 0.002 | 8.3% |
| Median | 0.016 | 0.018 | 0.002 | 13.8% |
| σ | 0.015 | 0.019 | 0.004 | 29.9% |
| MB | 0.000 | | | |
| RMSE | 0.021 | | | |
| MAD | 0.016 | | | |

[Figure]

Figure R1. Comparison of the observed and simulated concentrations of (a) total (gas + particulate) CHR and (b) total BaP at Beijing. The data at Beijing were daily, March-April 2005. Simulated results of RAQM2-POP model and WRF/Chem-PAH model are from Inomata et al. (2012) and this study, respectively.

**References**

Friedman, C. L., and Selin, N. E.: Long-Range Atmospheric Transport of Polycyclic Aromatic Hydrocarbons: A Global 3-D Model Analysis Including Evaluation of Arctic Sources, Environ. Sci. Technol., 46, 9501-9510, 10.1021/Es301904d, 2012.

Friedman, C. L., Pierce, J. R., and Selin, N. E.: Assessing the Influence of Secondary Organic versus Primary Carbonaceous Aerosols on Long-Range Atmospheric Polycyclic Aromatic Hydrocarbon Transport, Environ. Sci. Technol., 48, 3293-3302, Doi 10.1021/Es405219r, 2014.

Galarneau, E., Bidleman, T. F. and Blanchard, P. Seasonality and interspecies differences in particle/gas partitioning of PAHs observed by the Integrated Atmospheric Deposition Network (IADN). Atmos. Environ. 40, 182-197, 2006.

Hung, H., Kallenborn, R., Breivik, K., Su, Y., Brorström-Lundén, E., Olafsdottir, K., Thorlacius, J. M., Leppänen, S., Bossi, R., Skov, H., Manø, S., Patton, G. W., Stern, G., Sverko, E., and Fellin, P.: Atmospheric monitoring of organic pollutants in the Arctic under the Arctic Monitoring and Assessment Programme (AMAP): 1993–2006, Sci. Total Environ., 408, 2854-2873, 2010.

Inomata, Y., Kajino, M., Sato, K., Ohara, T., Kurokawa, J. I., Ueda, H., Tang, N., Hayakawa, K., Ohizumi, T., and Akimoto, H.: Emission and atmospheric transport of particulate PAHs in Northeast Asia, Environ. Sci. Technol., 46, 4941-4949, 10.1021/Es300391w, 2012.

Inomata, Y., Kajino, M., Sato, K., Ohara, T., Kurokawa, J., Ueda, H., Tang, N., Hayakawa, K., Ohizumi, T., and Akimoto, H.: Source contribution analysis of surface particulate polycyclic aromatic hydrocarbon concentrations in northeastern Asia by source-receptor relationships, Environ. Pollut., 182, 324-334, 10.1016/j.envpol.2013.07.020, 2013.

Kahan, T. F., Kwamena, N. O. A., and Donaldson, D. J.: Heterogeneous ozonation kinetics of polycyclic aromatic hydrocarbons on organic films, Atmos. Environ., 40, 3448-3459, 10.1016/j.atmosenv.2006.02.004, 2006.

Kim, J. Y., Ghim, Y. S., Song, C. H., Yoon, S. C., and Han, J. S.: Seasonal characteristics of air masses arriving at Gosan, Korea, using fine particle measurements between November 2001 and August 2003, J. Geophys. Res.-Atmos., 112, 10.1029/2005jd006946, 2007.

Kim, J. Y., Lee, J. Y., Choi, S. D., Kim, Y. P., and Ghim, Y. S.: Gaseous and particulate polycyclic aromatic hydrocarbons at the Gosan background site in East Asia, Atmos. Environ., 49, 311-319, 10.1016/j.atmosenv.2011.11.029, 2012.

Kwamena, N. O. A., Thornton, J. A., and Abbatt, J. P. D.: Kinetics of surface-bound benzo[a]pyrene and ozone on solid organic and salt aerosols, J. Phys. Chem. A, 108, 11626-11634, 10.1021/Jp046161x, 2004.

Lammel, G., Sehili, A. M., Bond, T. C., Feichter, J., and Grassl, H.: Gas/particle partitioning and global distribution of polycyclic aromatic hydrocarbons - A modelling approach, Chemosphere, 76, 98-106, 10.1016/j.chemosphere.2009.02.017, 2009.

Pöschl, U., Letzel, T., Schauer, C., and Niessner, R.: Interaction of ozone and water vapor with spark discharge soot aerosol particles coated with benzo[a]pyrene: O-3 and H2O adsorption, benzo[a]pyrene degradation, and atmospheric implications, J. Phys. Chem. A, 105, 4029-4041, 10.1021/Jp004137n, 2001.

Sehili, A. M., and Lammel, G.: Global fate and distribution of polycyclic aromatic hydrocarbons emitted from Europe and Russia, Atmos. Environ., 41, 8301-8315, 10.1016/j.almosenv.2007.06.050, 2007.

Shen, H. Z., Tao, S., Liu, J. F., Huang, Y., Chen, H., Li, W., Zhang, Y. Y., Chen, Y. C., Su, S., Lin, N., Xu, Y. Y., Li, B. G., Wang, X. L., and Liu, W. X.: Global lung cancer risk from PAH exposure highly depends on emission sources and individual susceptibility, Sci. Rep., 4, 10.1038/Srep06561, 2014.

Shrivastava, M., Lou, S., Zelenyuk, A., Easter, R. C., Corley, R. A., Thrall, B. D., Rasch, P. J., Fast, J. D., Massey Simonich, S. L., Shen, H., and Tao, S.: Global long-range transport and lung cancer risk from polycyclic aromatic hydrocarbons shielded by coatings of organic aerosol, Proceedings of the National Academy of Sciences, 10.1073/pnas.1618475114, 2017.

Torseth, K., Aas, W., Breivik, K., Fjaeraa, A. M., Fiebig, M., Hjellbrekke, A. G., Myhre, C. L., Solberg, S., and Yttri, K. E.: Introduction to the European Monitoring and Evaluation Programme (EMEP) and observed atmospheric composition change during 1972-2009, Atmos. Chem. Phys., 12, 5447-5481, 10.5194/acp-12-5447-2012, 2012.

Zhang, Y., Tao, S., Ma, J., and Simonich, S.: Transpacific transport of benzo[a]pyrene emitted from Asia, Atmos. Chem. Phys., 11, 11993-12006, 10.5194/acp-11-11993-2011, 2011a.

Zhang, Y. X., Shen, H. Z., Tao, S., and Ma, J. M.: Modeling the atmospheric transport and outflow of polycyclic aromatic hydrocarbons emitted from China, Atmos. Environ., 45, 2820-2827, 10.1016/j.atmosenv.2011.03.006, 2011b.

---

## Author Comment (AC2) · 23 Aug 2017

**Response to comments of referee #2**

**General comments:**

PAHs are substances the can severely impair human health and ecosystems. However, still little is known about their fate in the environment, in particular in the atmospheric environment. By investigating the atmospheric transport of some important PAHs with the state-of-the art model WRF/Chem in China, a region where the pollution by PAHs is supposed to be significant, the authors provide an important contribution to PAH research. The physical-chemical processes they implement into WRF/CHEM are based on sound science. Also, the general setup of the model study appears sound to me.

**Response:**

   Many thanks for the encouraging words. Please kindly find our point-to-point response to your questions/comments below.

1. However, I'm not convinced that the authors treat the PAH emissions - a crucial part of modeling studies - in an appropriate way. They use emissions of 2008 and compare model results to measurements of 2013 and 2003. This only makes sense if PAH emissions did not change within this period. On the other hand, the authors convincingly explain the increasing relevance of PAHs in Asia due to rapidly increasing emissions. If the authors applied inter-annual scaling factors for modeling the years where the measurements were carried out, they should explain in detail how this was done. Otherwise, they should comment on this contradiction. An agreement with measurements alone could also be "right for the wrong reason".

**Response:**

   Thanks for raising this concern. We indeed scaled the anthropogenic PAH emissions from the base year 2008 to the simulated years 2003 and 2013, with scaling factors explained in page 8 line 4. The inter-annual scaling factors are taken from Shen et al. (2013), the same paper that introduces the global PAH emission inventory in 2008 used in our study. In this paper, historical time trends (e.g. our modeling year 2003) are based on the historical fuel consumption data and time-dependent emission factor of PAH. Future time trends (e.g. our modeling year 2013) are predicted using the IPCC SRES A1 scenario supposing a future world of rapid economic growth (Nakićenović et al. 2000). Besides, we have also applied monthly scaling factors following Zhang and Tao (2008). For clarity, we add a new Figure R1 (SI Figure S3) to illustrate the scaling of PAH emissions and rephrase the sentence in page 8 line 4 into *"For specific simulation period, inter-annual scaling factors in the simulated domain are taken from Shen et al. (2013) based on historical fuel consumption data and IPCC SRES A1 scenario supposing a future world of rapid economic growth. Monthly scaling factors are taken from Zhang and Tao (2008)"*.

[Figure]

Figure R1 (SI Figure S3). (a) Inter-annual, (b) monthly and (c) hourly scaling factors for PAH emissions.

2. For evaluating the model against measurements the authors compare arithmetic means and Pearson's correlation coefficients of time series, I assume (they don't mention it explicitly). In figure 3 and 5 one can see that the error bars reach negative values. This indicates non-normal distributions and statistical measures like arithmetic mean and standard deviations cannot be applied. The authors must check the distribution and decide based on this which measures to use.

**Response:**

The correlation coefficients shown in the figures use combined daytime and nighttime data sets, i.e., between 24 observation and 24 simulation samples (12 daytime and 12 nighttime). These correlation coefficients have all passed the Student's t-test with a significance level of 0.05. Correlation coefficients for only daytime or nighttime samples are not included, because of small

data sets. In Fig. 5, correlation coefficients are also not included for the same reason. We have added the clarifications of correlation coefficients into the caption of Fig. 3: *The correlation coefficients R use combined daytime and nighttime data sets, passing the Student's t-test with a significance level of 0.05.*

In Fig. 3 and Fig. 5, some of the error bars reach negative values because the error bars indicate the standard deviations which are larger than the arithmetic means. This implies that the data is widely spread out to the mean. The same applies for the observed and simulated PAH data in another similar modeling study applied to this region (Inomata et al., 2012). Although the concept of standard deviation and arithmetic mean are not limited to normally distributed data, we do admit that two statistic metrics are not enough, so that more metrics are now provided in the Table R1 (SI Table S3) and Table R2 (SI Table S4) to characterize model performance in the Xianghe summer case (Fig. 3) and the Gosan winter case (Fig. 5), respectively.

Table S1 (SI Table S3). Observed (obs) and simulated (sim) mean concentration, median, standard deviation ($\sigma$), mean bias (MB), root mean square error (RMSE), mean absolute deviation (MAD) in unit ng m$^{-3}$ and correlation coefficient (R) at the Xianghe site averaged over 11–22 July, 2013. The correlation coefficients use combined daytime and nighttime data sets, passing the Student's t-test with a significance level of 0.05.

**gaseous PHE**

| | daily | | | | day | | | | night | | | |
| --- | --- | --- | --- | --- | --- | --- | --- | --- | --- | --- | --- | --- |
| | obs | sim | sim-obs | (sim-obs)/obs | obs | sim | sim-obs | (sim-obs)/obs | obs | sim | sim-obs | (sim-obs)/obs |
| Mean | 25.6 | 22.2 | -3.4 | -13.4% | 13.6 | 11.7 | -1.9 | -14.1% | 36.5 | 32.6 | -3.9 | -10.7% |
| Median | 23.9 | 15.5 | -8.5 | -35.5% | 13.0 | 12.5 | -0.5 | -4.2% | 33.3 | 35.3 | 2.0 | 6.0% |
| σ | 13.8 | 14.8 | 0.9 | 6.7% | 5.1 | 5.2 | 0.1 | 2.5% | 9.6 | 13.8 | 4.2 | 43.8% |
| MB | -3.0 | | | | -2.0 | | | | -3.9 | | | |
| RMSE | 10.5 | | | | 5.4 | | | | 13.6 | | | |
| MAD | 7.9 | | | | 4.5 | | | | 11.1 | | | |
| R | 0.72 | | | | | | | | | | | |

**gaseous CHR**

| | daily | | | | day | | | | night | | | |
| --- | --- | --- | --- | --- | --- | --- | --- | --- | --- | --- | --- | --- |
| | obs | sim | sim-obs | (sim-obs)/obs | obs | sim | sim-obs | (sim-obs)/obs | obs | sim | sim-obs | (sim-obs)/obs |
| Mean | 0.63 | 1.27 | 0.65 | 102.8% | 0.48 | 0.65 | 0.17 | 34.5% | 0.76 | 1.90 | 1.14 | 149.2% |
| Median | 0.56 | 1.08 | 0.52 | 94.0% | 0.42 | 0.61 | 0.19 | 45.5% | 0.81 | 1.98 | 1.17 | 143.8% |
| σ | 0.28 | 0.77 | 0.49 | 170.9% | 0.19 | 0.27 | 0.08 | 42.6% | 0.29 | 0.57 | 0.28 | 97.5% |
| MB | 0.67 | | | | 0.16 | | | | 1.14 | | | |
| RMSE | 0.97 | | | | 0.35 | | | | 1.31 | | | |
| MAD | 0.74 | | | | 0.30 | | | | 1.14 | | | |
| R | 0.42 | | | | | | | | | | | |

**particulate CHR**

| | daily | | | | day | | | | night | | | |
| --- | --- | --- | --- | --- | --- | --- | --- | --- | --- | --- | --- | --- |
| | obs | sim | sim-obs | (sim-obs)/obs | obs | sim | sim-obs | (sim-obs)/obs | obs | sim | sim-obs | (sim-obs)/obs |
| Mean | 0.85 | 2.46 | 1.61 | 189.6% | 0.78 | 0.52 | -0.26 | -33.4% | 0.92 | 4.40 | 3.48 | 378.0% |
| Median | 0.58 | 0.98 | 0.40 | 69.1% | 0.45 | 0.45 | 0.00 | 0.7% | 0.97 | 4.89 | 3.92 | 404.4% |
| σ | 0.75 | 2.65 | 1.90 | 252.7% | 0.99 | 0.35 | -0.64 | -64.7% | 0.37 | 2.52 | 2.15 | 579.3% |

| | | | | | | | | | | | | |
|---|---|---|---|---|---|---|---|---|---|---|---|---|
| MB | 1.83 | | | | 0.03 | | | | 3.48 | | | |
| RMSE | 3.07 | | | | 0.46 | | | | 4.23 | | | |
| MAD | 2.00 | | | | 0.35 | | | | 3.51 | | | |
| R | 0.59 | | | | | | | | | | | |

| particulate BaP | | | | | | | | | | | | |
|---|---|---|---|---|---|---|---|---|---|---|---|---|
| | daily | | | | day | | | | night | | | |
| | obs | sim | sim-obs | (sim-obs)/obs | obs | sim | sim-obs | (sim-obs)/obs | obs | sim | sim-obs | (sim-obs)/obs |
| Mean | 0.78 | 0.78 | 0.00 | 0.3% | 0.43 | 0.14 | -0.29 | -68.1% | 1.10 | 1.37 | 0.27 | 24.5% |
| Median | 0.49 | 0.29 | -0.20 | -41.1% | 0.39 | 0.13 | -0.26 | -67.1% | 1.50 | 0.89 | -0.61 | -40.6% |
| σ | 0.81 | 0.70 | -0.12 | -14.5% | 0.24 | 0.08 | -0.16 | -67.5% | 0.81 | 0.73 | -0.08 | -10.1% |
| MB | 0.002 | | | | -0.29 | | | | 0.27 | | | |
| RMSE | 0.61 | | | | 0.38 | | | | 0.76 | | | |
| NMB | 0.48 | | | | 0.31 | | | | 0.63 | | | |
| R | 0.69 | | | | | | | | | | | |

Table R2 (SI Table S4). Same as SI Table S3 but at the Gosan site averaged over 14–25 February, 2003.

| gaseous PHE | | | | |
| --- | --- | --- | --- | --- |
| | obs | sim | sim-obs | (sim-obs)/obs |
| Mean | 0.81 | 1.73 | 0.92 | 113.6% |
| Median | 0.54 | 1.14 | 0.60 | 109.8% |
| σ | 0.57 | 1.95 | 1.38 | 242.1% |
| MB | 0.92 | | | |
| RMSE | 2.35 | | | |
| MAD | 1.32 | | | |

| gaseous CHR | | | | |
| --- | --- | --- | --- | --- |
| | obs | sim | sim-obs | (sim-obs)/obs |
| Mean | 0.03 | 0.03 | 0.000520 | 1.8% |
| Median | 0.02 | 0.02 | 0.00 | -4.6% |
| σ | 0.02 | 0.03 | 0.01 | 69.8% |
| MB | 0.00 | | | |
| RMSE | 0.04 | | | |
| MAD | 0.03 | | | |

| particulate CHR | | | | |
| --- | --- | --- | --- | --- |
| | obs | sim | sim-obs | (sim-obs)/obs |
| Mean | 0.45 | 0.24 | -0.21 | -47.5% |
| Median | 0.40 | 0.06 | -0.34 | -84.1% |
| σ | 0.35 | 0.44 | 0.09 | 26.2% |
| MB | -0.21 | | | |
| RMSE | 0.51 | | | |
| MAD | 0.36 | | | |

| particulate BaP | | | | |
| --- | --- | --- | --- | --- |
| | obs | sim | sim-obs | (sim-obs)/obs |
| Mean | 0.020 | 0.022 | 0.002 | 8.3% |
| Median | 0.016 | 0.018 | 0.002 | 13.8% |
| σ | 0.015 | 0.019 | 0.004 | 29.9% |
| MB | 0.000 | | | |
| RMSE | 0.021 | | | |
| MAD | 0.016 | | | |

3. The authors use expressions like "good prediction", "fair agreement", "significantly improved", ... to judge their model results. They should explain by which criteria they consider a result (an average or a correlation) as good or not good. As they didn't perform any statistical tests it seems to be pure opinion. I found only one statement where they explain their opinion: Compared with previous studies ... (page 10, line 4).

**Response:**

Thanks. We add more statistic metrics in Table R1 and Table R2 to characterize model performance in the Xianghe summer case (Fig. 3) and the Gosan winter case (Fig. 5), respectively. Also, we try to discuss model performance by comparing with previous global and regional model studies. For example, in "4.2.2 Evaluation of the Asian outflow" we have the following comparisons *"Model validation so far had been limited to seasonal features (Zhang et al., 2011a; Zhang et al., 2011b), while higher temporal features had not been addressed yet. For example, discrepancies of a factor of 16–476 between predicted and observed average PAH (BaP, CHR, BbF, BkF, IcdP, DahA, BghiP) concentrations at the Waliguan site, a continental background site for ambient air monitoring in western China, were found much larger than at urban or suburban sites (Zhang et al., 2009)"*. Another example in this section is *"Compared with previous studies, our simulated average concentrations of BaP agreed well with the observation (deviation < 10%), while Zhang et al. (2011a) underestimated BaP by about 50%. For the Gosan summer case, our simulated average BaP concentration is 0.006 ng m$^{-3}$ (Fig. S6), much closer to the observed 0.012 ng m$^{-3}$ than the simulated BaP concentration of $\approx$ 0.001 ng m$^{-3}$ by Zhang et al. (2011a)"*. In "4.2.1 Evaluation at the near source areas", we add comparisons of simulated daily results with one previous regional model evaluation in Beijing (Inomata et al., 2012), since Xianghe is a semi-urban town in the Beijing metropolitan area. Page 9 line 8, *"PAH diurnal variabilities are well captured for both gas- and particulate-phase species at the Xianghe site, with correlation coefficients of 0.42–0.72 (Fig. 3, Table S3) compared with 0.30–0.58 in Beijing (Xianghe is a semi-urban town in the Beijing metropolitan area) by Inomata et al. (2012)"*. Page 9 line 14, *"The model well catches the observed daily average concentration of particulate BaP (observed 0.78 ng m$^{-3}$, simulated 0.78 ng m$^{-3}$), while Inomata et al. (2012) underestimated daily concentration of BaP in Beijing by a factor of 2"*.

However, it is not easy to find proper criteria for daytime and nighttime concentrations as well as particulate mass fraction of PAH. Previous model evaluation is unavoidably quite limited due to rare and often incomplete (e.g. particulate phase only) monitoring activities (almost none in Eastern Asia) with low temporal resolution. The results shown in this study are beyond the context of previous PAH modeling studies. Considering that to predict the diurnal cycle and the mass fraction of PAH involves higher temporal resolution and more complex processes than to predict daily concentration, we find the model performance good enough since it could meet the similar standard as daily concentration. When more PAH monitoring data becomes available in the near future, more features of simulated results can be evaluated.

**Minor comments:**

1. page 1 line 29: To my knowledge the word "tracer" is used for inert substances (which PAHs are not). For substances in very low concentrations I would rather suggest to use "trace gases/compounds/substances".

**Response:**

Thanks. We intend to use the word "tracer" to include both inert and reactive substances in the modification of transport scheme, not only PAHs. In fact, the usage of "tracer" is not totally consistent throughout the literature, e.g., the widely used chemical mechanism Model for Ozone and Related Chemical Tracers version 4 (MOZART-4) uses "tracer" to represent reactive substances. Therefore, we have changed the word "tracer" into "species" to avoid any misunderstanding.

2. page 2 line 27ff: the information of item 4 is included in item 1 and could be left away. Figures 2c and 2d are not necessary because the authors explain in the text (page 6 line 32ff) why the transport behavior of trace substances is inherent to the model.

**Response:**

Thanks. Item 1 is to implement and introduce all the latest schemes in Section 2, but item 4 shows sensitivity tests to explain why some of the processes are indeed necessary in Section 5. Considering that item 4 proves the indispensability of new processes and contributes an independent section, it might be better to separate item 4 from item 1 as it is.

Figure 2c and 2d demonstrate an example of BC, a species that is already included in the WRF/Chem model, to exclude the possibility that our incorrect method would cause such abnormal transport behavior. The example of BC also proves that this transport problem occurs not only for PAH but also for any low concentrated species. Therefore, we tend to keep Fig. 2c and 2d for the above reasons.

References

Inomata, Y., Kajino, M., Sato, K., Ohara, T., Kurokawa, J. I., Ueda, H., Tang, N., Hayakawa, K., Ohizumi, T., and Akimoto, H.: Emission and atmospheric transport of particulate PAHs in Northeast Asia, Environ. Sci. Technol., 46, 4941-4949, Doi 10.1021/Es300391w, 2012.

Shen, H. Z., Huang, Y., Wang, R., Zhu, D., Li, W., Shen, G. F., Wang, B., Zhang, Y. Y., Chen, Y. C., Lu, Y., Chen, H., Li, T. C., Sun, K., Li, B. G., Liu, W. X., Liu, J. F., and Tao, S.: Global Atmospheric Emissions of Polycyclic Aromatic Hydrocarbons from 1960 to 2008 and Future Predictions, Environ. Sci. Technol., 47, 6415-6424, Doi 10.1021/Es400857z, 2013.

Zhang, Y. X., and Tao, S.: Seasonal variation of polycyclic aromatic hydrocarbons (PAHs) emissions in China, Environ. Pollut., 156, 657-663, DOI 10.1016/j.envpol.2008.06.017, 2008.

Nakićenović, N., et al. Emissions Scenarios: A Special Report of Working Group III of the Inter-Governmental Panel on Climate Change, Cambridge University Press, New York, 2000.